# Developing the hertz art-science project to allow inaudible sounds of the Earth and Cosmos to be experienced

Graeme J. Marlton[1], Juliet Robson[2]

[1]Department of Meteorology, University of Reading, Reading, UK, RG6 6BB
[2]Wyfold Lane Studio, Wyfold Lane, Peppard, RG9 5LR

*Correspondence to*: Graeme.marlton@reading.ac.uk

**Abstract.** The Earth and atmosphere are in constant motion. Volcanoes, Glaciers, Earthquakes, Thunderstorms and even the Aurora produce powerful low frequency sounds known as Infrasound. Infrasound is constantly passing through our atmosphere at frequencies less than 20 Hz, below the range of human hearing, effectively an inaudible symphony. Inspired by wanting to allow physical access to this natural phenomenon, a collaboration between the worlds of contemporary art and meteorology has been developed. This led to a project called **hertz,** named after the nineteenth century physicist Henirich Hertz whose surname provides the scientific unit (Hz) for frequency. **hertz** explores the manifestation of the hidden vibrations of our own planet and the secret harmonies of our stars. The manifestation of hidden vibrations of our own planet was principally achieved using a subwoofer and furniture adapted to vibrate to the amplitude of infrasonic waves from pre-recorded sources and in real time. The project's motivations are in exploring new methods to experience and re-engage with parts of our planet through this phenomenon. **hertz** has had a UK national tour in which seven thousand people interacted with the piece, of which approximately 85% felt more reconnected to the environment after interacting with the installation. This paper describes the concepts, creative ideas, technology, and science behind the project. It addresses its development, including the steps to make it accessible for all, and examines its impact on those who created and interacted with the work.

## 1 Introduction

The Earth and atmosphere are in constant motion due to a range of natural processes such as seismic activity, volcanic eruptions, and glacial slippage. Atmospheric phenomena such as hurricanes, thunderstorms and tornadoes also contribute. At first-hand, these events can be both majestic and alarming. Increasingly, these are rarely experienced directly, as more of the Earth's population lives in towns and cities, insulated from these expressions of nature. The acoustic signals of natural terrestrial and atmospheric changes are evermore obscured by the background anthropogenic noise of airports, trains, and motorways. Technology further isolates the modern human from the natural environment in which we evolved. Seeking to re-invigorate and inspire our relationship with the natural environment through the use of inaudible frequencies. One of us, the artist Juliet Robson, aimed to create an interactive art work, that would re-establish this diminishing link, one that was tangible in a very real sense and that allowed a way into the important but sometimes inaccessible research done by scientists. This

would align with the view that "Artists are no longer concerned with creating artwork that reflects or interprets reality; rather, they want to be active agents in creating it, ... That means that artists need to have an even deeper understanding of the mechanics behind science and technology.'' (Williams, 2017).

To undertake this, it was apparent that such a project would need to call on science, technology, engineering, and maths (STEM) expertise to create an authentic as possible representation of natural hidden vibrations through an immersive experience. Robson approached two scientists and a mathematician to explore the possibilities of making hidden frequencies of the stars and natural phenomena of our planet heard and felt. Art-science collaborations highlighting unseen and intangible processes occurring around us and demonstrated to the wider public have been undertaken before and continue to generate interest. Ezquerro et al. (2019) sonified sediment samples to make 14 distinct compositions based on the lithography and structure of each sample. Hooker (2011) designed an installation which detected high energy particles from outside our solar system, which pass through humans without leaving a trace, and used it to trigger notes on an electric piano. Collaborating with scientists and researchers globally, Patterson (2007) created a phone line that could be called from anywhere in the world through which you could listen to a glacier melting. In addition to this McMullen (2005) described how a group of artists worked with scientists at CERN to create artworks which reflected theories in the realm of physics such as the crumple effect.

One of the scientists Robson contacted was, co-author, Graeme Marlton, a meteorologist who was working on the Atmospheric Research Infrastructure in Europe 2 (ARISE 2) project (Blanc et al. 2018). The project encompassed examining a multitude of different novel measurement techniques to measure the dynamical properties of the atmosphere. One such technology utilised in ARISE2 was infrasound measurements. Infrasound contains sound frequencies which fall below the audible range of human hearing, essentially sound waves below 20 Hz. It is produced naturally, or artificially by large explosions such as that of a nuclear detonation or by mining activity, as well as trains and planes. Natural infrasound is produced by volcanoes, earthquakes, glaciers, ocean swell, thunderstorms, hurricanes and even the aurora borealis (Wilson 1969) as shown in figure 1. The importance of infrasound to the ARISE project was to learn about the state of the atmosphere by learning how infrasonic waves passed through it from a known infrasound source, such as a volcano (Smets et al 2019).

It was suggested by Marlton that infrasound could be used as a medium for Robson's new project and Robson was interested in the possibility of experiencing its inaudible symphony. Infrasound has featured in art installations before, Grachrow (2005) produced an installation called the Long wave synthesis which aimed to challenge how we perceive our environment and long wave vibration. Grupfinger (2009) built an installation that allowed the audience to experiment and experience a range of infrasound. Anish Kapoor (Aerotrope 2012) produced an installation in collaboration with an engineering company that played infrasound through the human body in a confined place. This was not Kapoor's only work using infrasound, Barres (2017) discussed in a review of Kapoor's works how Infrasound was played at 18 Hz in a room to make the room feel haunted. He was not successful, the installation caused anxiety amongst the visitors and museum staff.

To provide new access to natural infrasound, the raw infrasound data could be processed to provide a sound wave which could be played through commercially available transducers. Transducers are devices which shake when a low frequency sound is played through them. They are fitted in 4D cinemas and video gaming chairs to provide a vibrating sensation to the spectator when an explosion or aircraft passes over on screen to make the experience more immersive. The aim here was to enable people to feel their bodies resonating to the inaudible symphony of the planet. Figure 2 shows the initial concept work drawn

up for the project. This would become one of the two working strands for Robson's immersive new project.

The other strand is not discussed at length here, but a brief description is given here. It took astronomical measurements of the stars and from their spectral signature derive a sound wave. Sound waves move through a star's gaseous interior because of temperature changes which cause the star to fluctuate in brightness. Satellites such as the North American Space Agency

(NASA) 's Kepler satellite and the Transiting Exoplanet Survey Satellite, can observe these vibrations. Data for hertz was obtained from NASA's Kepler project (Chaplin et al 2010) and then sonified and played through Chladni plates built for the project. Chladni plates consist of a flat sheet of metal, usually circular or square, mounted on a central stalk to a sturdy base. When the plate oscillates at a mode of vibration, the nodes, and antinodes form complex but symmetrical patterns over its surface. The positions of these nodes and antinodes can be seen by sprinkling sand upon the plates, the sand will vibrate away

from the antinodes and gather at the nodes (Stöckmann 2007). As the frequencies of different stars were played through the plates, the sand sprinkled on the plate formed geometric patterns related to that star's frequency. Chladni plates have been used as a method of visualising sounds and music as documented by Stanford (2013).

As the connection between the two strands was that of vibrations, oscillations and resonance which are associated with

85 frequency, the project was named **hertz,** after the standard unit of frequency. Drawing on the fact that everything vibrates, from the smallest atom to the furthest star, their frequencies surround us and yet leave no imprint, **hertz** enables people to feel their bodies resonating to the inaudible symphony of our own planet, experience the stars singing and see their sound made visible. **hertz**'s ultimate goal would aim to reconnect us to our planet and place in the cosmos. Its ancillary aims would also be to educate about the science behind the project.

In this paper we predominantly focus on the infrasound strand of the **hertz** project. In section 2 we will describe the science behind how the installation works and the initial feedback received on the prototype. In section 3 we describe how the feedback modelled the version prepared for the **hertz** tour around the UK. Section 4 discusses accessibility considerations for the project and tour. Section 5 discusses the **hertz** set up at each of the three UK tour locations. In section 6 we review the feedback from

95 the public from the tour. In section 7 we discuss the collaboration from between the artist and the scientist. The project findings are summarised in the conclusions in section 8.

**2 hertz from concept to prototype**

To create an immersive experience where modified infrasound is played through a transducer, infrasound recordings which had captured the acoustics of the natural world were needed. In this section we describe how infrasound is measured and how the infrasonic recordings used in the **hertz** project were acquired. We will then describe the prototype setup and how the infrasound recordings were processed to create an immersive experience.

**2.1 Infrasound recordings**

Infrasound cannot be detected using normal audio recording equipment. Instead, a microbarometer, a very sensitive pressure sensor, can be used to detect the subtle pressure variations generated by infrasound. Globally, networks of microbarometers are maintained by meteorological and seismological organisations. The sensor used for **hertz** is from a US based company, InfilTech, manufactures a small portable, low-cost infrasound detector, the INFRA20, which can be logged via a serial port to a standard computer. It was initially deployed in the suburbs of Reading during where it measured infrasound from several thunderstorms that had formed over northern France and had moved northwards over the English Channel and into southern England on the 18th July 2017. Figure 3 shows a spectrogram - an image that displays the detected infrasonic frequencies - for these events. The spectrogram has a horizontal time axis, and frequency on the vertical axis, using colour to indicate the amplitude of the signal. Figure 3 shows as each thunderstorm approached infrasound frequencies in the 0.02 Hz to 1 Hz range were generated. The most intense frequencies were detected from the thunderstorms at 3 Local Time (LT). From 7LT to 9LT there are different low amplitude infrasonic waves detected in the 0.5 to 1 Hz range, likely to be associated with traffic and trains during the morning rush hour. To contextualise this, the infrasonic waves observed here oscillate once over a period of 10 seconds whereas the sound from a subwoofer will oscillate over a period of a hundredth of a second.

The INFRA20 was also used to record the infrasonic signal from the aurora borealis at Pallas, Northern Finland in September 2017. The infrasound there had a distinct signature below 1 Hz as shown in figure 4 and in agreement with (Wilson 1969). The scaling of the colour bar in the periodograms shows that the amplitude of the infrasonic signals produced by the aurora is four times smaller than the amplitude of the infrasonic waves of the thunderstorms. Furthermore, the infrasonic signals produced by the aurora occupy a much lower frequency range than the thunderstorms. This shows that different phenomena produce different infrasound signatures. In addition to the recordings made directly with the INFRA20, infrasound data clips were also provided by ARISE2 project members. These included infrasonic recordings of Mount Etna, and an F16 jet aircraft accelerating to speeds greater than that of sound.

**2.2 Hertz prototype test rig**

Figure 2 showed a concept picture for the infrasound setup. For prototyping, a setup shown in figure 5 using a large subwoofer loudspeaker (250 Watt) and an ADX maximus transducer was implemented. The transducer had a clamp allowing it to be attached to a chair or wheelchair, which quickly became the furniture of choice for prototyping. Robson had a spare metal

wheelchair that was good at transferring vibrations which boasted a variety of possible mounting points where the transducer could be attached. In addition, it was easy to move the wheelchair to different areas of the studio to experience and experiment with different spatial configurations. Both the transducer and subwoofer were connected to the soundcard of a computer meaning the same processed infrasound signal could be played through both simultaneously. A subwoofer that could play low frequency sounds down to 60 Hz was used to increase the immersive experience and so that audiences could be attracted to

the installation from a distance. This also stimulated another sense, hearing, by providing loud deep sounds complementing the vibrations provided by the transducer that provided access through physical sensations in the body. The transducer was designed to only play audio signals between 120 Hz and 40 Hz. Given the power delivered through the subwoofer and transducer, the opening track of Pink Floyd's *Dark side of the Moon* (Pink Floyd 1973), played merely as a test track, led to tremendous shaking of the modified chair and studio in which it was placed.

**2.3 Infrasound processing method**

The next part of the project was to turn the infrasonic recordings described in section 2.1 into something that could be played through the transducer and large subwoofer. In their current state they would be inaudible and would not register on the transducer or subwoofer. In addition to this background noise, for example from wind passing over the sensor, also needed to be filtered out. To achieve this a digital bandpass filter was applied over the raw infrasound data. A bandpass filter is a physical

or software device which allows a frequency between two given frequencies to pass, whilst frequencies outside of this range are removed. The spectrograms in figures 2 and 3 were used to define the upper and lower limits of the band pass filter, by establishing the frequency range in which the infrasonic signatures were largest.

The first approach was to use the amplitude of the bandpass filtered infrasound signal to modulate a tone at a range of low

frequencies between 60 and 100 Hz. To achieve this the infrasonic time series was first band pass filtered to yield *BP(t)* and was then multiplied element wise by a sine wave of given frequency *f* to give a sound wave

$$X(t) = BP(t)\sin(2\pi f t) \ (1)$$

where *t* is the time index. This gave mixed results. At first it gave an unworldly noise, with the rig making a zooming noise as the shaking and rumbling changed intensity at random speeds, sounding like a sci-fi effect. A single tone was successful in

yielding an interpretation of infrasound. However, we felt that it did not encapsulate what infrasound might sound like if we could hear it. One thing which was lacking was a depth, which was largely due to the monochromatic tone used and it was felt that a mix of frequencies would amount to a larger sense of resonant layers and feeling of being immersed in the infrasound. Hence, an alternative was to create a deep cacophony of tones. The method to achieve this was to firstly create pink noise. Noise is sometimes described by likening its spectrum to the optical spectrum of colours. White noise is the hiss noticeable on

radios tuned away from a radio station, and its spectral power is constant over all frequency bands. Pink noise's spectral power is inversely proportional to the audio frequency. This gives an effect where low frequency noise is more dominant than higher frequency noise, giving a rumbling sensation that surrounds and is felt bodily like sitting on an airplane.

The bandpass filtered infrasound signal was then used to modulate the amplitude of synthetic generated pink noise. To ensure a deep rumbling was experienced through the prototype rig a further low pass filter, a filter similar to the band pass filter but only removing high frequency sounds was then applied. This produced a low rumbling noise to be played through both the subwoofer and transducer, the rumbling changing in amplitude as determined by the raw infrasonic signal. This produced an effect that we felt was relatable to infrasound if we could hear it, while keeping translatable authenticity, something that was important to **hertz**'s ethos. This pink noise based processed infrasound recordings now had more depth and independent character depending on the infrasound clip used which began conveying an emotion and sense of majesty about our planet that the project had yearned to create.

As part of the development process some of the initial testing was videoed using a smartphone. However, as discussed earlier, low frequencies cannot be detected through conventional sound recording equipment. Thus, on playback through mobile phone or computer, the modulated infrasound was inaudible and only the vocal reactions and the rattling of loose objects on tables were audible. A video example from the development phase can be found here https://researchdata.reading.ac.uk/id/eprint/267

This meant you had to physically be present in order to sense the vibrations. Therefore, making **hertz** immersive and experiential, the changes in air pressure caused by the subwoofer can be felt in the space and in your body. The visceral and audible nature of the experience cannot be documented and played back. The infrasound generated is unique to the place it was recorded in and that moment cannot be replicated. This is one of the aspects that differentiates **hertz** from other artworks that use infrasound.

For the interested reader audio clips where the low pass filter was set to 300 Hz can be found at https://researchdata.reading.ac.uk/id/eprint/267 which were used on BBC Radio 3's late Junction with Max Reinhardt. In addition to this a technical appendix has been created that included in some further detail the filter coefficients and the equipment used.

### 2.4 Outreach activities and reception

Initial development of the prototype rig finished in late 2017. Following this several opportunities arose to demonstrate the prototype rig to the public and experts in both art and science fields. Table 1 shows a list of public outreach events. The largest of these events was the "Be there at the start" Conference hosted at the Attenborough Centre, a contemporary art centre and gallery located in Leicester, United Kingdom. The conference was organised by the project's funders, Unlimited, who facilitate new work by disabled artists to reach national and international audiences. Attendees had a wide range of disabilities, figures 6 and 7 show people with visual and hearing impairments respectively interacting with the prototype. Accessibility considerations for **hertz** are considered in section 4.

One of the key questions was to find out what people thought of the artwork and if the experience was uncomfortable and not enjoyable. Responses were sought from those experiencing the artwork, which included:

- 'Epic.' 'Ground-breaking.' 'A whole world around me I couldn't see but felt connected to.'
- 'In this piece, I can time travel and contemplate the geometry of sound into matter – wow. Mind-blowing and poetic.'
- 'Incredible vibrations.
- 'Primeval, dramatic, disconcerting and yet thrilling.'

While feedback gained was through verbal communication and written comments, the majority was of an enjoyable or interesting experience. There were no negative comments in terms of discomfort in the written feedback comments, so it is difficult to know whether anybody found the experience uncomfortable. The words 'intense' and 'soothing' were used in conversation by visitors. It is possible that if someone found the experience unpleasant, they left the room and did not comment. But that was not ascertained. Robson experiences chronic pain and had found no ill effects, some vibrations were soothing and some intense which did not exacerbate her chronic pain. It should be noted that participation was voluntary, and any participant could leave when they wanted.

In addition to the feedback received in person, Max Reinhardt of BBC Radio 3's late Junction played excerpts of infrasound from the aurora borealis. The infrasound had been reprocessed, so it was just audible for listening on radio on his show and said: 'What a totally astounding and amazing project'. There was further positive media coverage of the initial prototyping in Disability Arts Online Magazine (Caulfield 2017) which quotes: "**hertz**, promises to redefine the boundaries of our perception of the stars and the nature of sound." and later states that: "... the **hertz** team is practically a work of art itself" . Kalaugher (2018) who visited **hertz** at the European Geophysical Union conference, Vienna, Austria, wrote 'It's not every day you get shaken by Etna'.

**3 Developing hertz for tour**

Following the running of the prototypes at the venues shown in table 1, and the positive feedback from the public, a tour was commissioned which would see **hertz** being exhibited to the public at three places across the UK.  It was realised if the installations were to tour, further development would be needed. The first extensive upgrade was to increase the amount of furniture that vibrated allowing more people to experience the infrasound vibrations. The second was to upgrade the software that played the infrasound through the subwoofer and transducers. This was to make it (a) stand-alone, meaning minimal operator input, and (b) configuring the software to play infrasound recorded at the locale of installation in real-time. The first part of the work was to replace the wheelchair and attached transducer with more rigid furniture. A steel garden bench and chair which conducted vibrations well were each fitted with a transducer and linked to the existing subwoofer and playback

system. The second part, to overhaul the playback system, involved replacing the laptop PC shown in figure 5, with a small

stand-alone computer (a Raspberry Pi) so it could be easily concealed. The INFRA20 infrasound sensor's cable was extended so it could be placed outside whilst being connected to the Raspberry Pi. Further to this the Pi was configured to obtain data from the infrasound sensor, process it, and play back the processed infrasound signal in real time through the subwoofer and transducers. This allowed the real time infrasound of a location to be experienced. As computer peripherals such as a mouse, keyboard and monitor would detach from the aesthetics of the installation, the Pi was configured to run in a 'dead head' mode,

meaning a graphics user interface was not needed and any settings could be altered solely through keyboard commands. The Pi was also configured to begin the real time acquisition of data on start up further minimizing operator input.

## 4 Accessibility considerations

It was important to all involved in the project that accessibility was incorporated where possible from the start, particularly

regarding physical access and interpretation of **hertz**. From the outset accessibility considerations such as wheelchair access to buildings at the University of Reading allowed Robson to have initial discussions with Marlton. Research, development, and construction inevitably needed to be done in accessible venues and was done at Robson's studio, the University of Reading and at 101 Outdoor Arts Creation Space, Newbury, where the finalised version of **hertz** was constructed.

The furniture used to transmit the infrasound vibrations of the location in real time for the tour was chosen not only for its conductive qualities and ability to be used outside but for its sturdiness. The highest seat possible of this type of furniture was sourced allowing it to be sat on easily. In addition to the two benches a chair with arms was used for visitors who needed more support. During the development phase the transducers could be transferred to another participant's wheelchair to allow them to partake in the experience without the need to leave their chair. The Volume of the sound emitted from the sub-woofer was

set relatively high for impact and so that visitors could 'feel' it in the air pressure changes and in their bodies. Care was taken not to exceed each venue's health and safety guidelines. The subwoofer was placed in each venue where it was easy to touch and get close to, so that visitors could feel the vibrations and feel the gusts of air pumped out by the speaker as it amplified the sound. Figure 7 shows a deaf visitor touching the subwoofer so they could experience the processed infrasound.

Physically, all venues used during the tour and described in section 5 had the minimum of wheelchair access with disabled toilets and lifts. We The Curious, where **hertz** was installed for three months, had comprehensive access with detailed information on their website including a virtual accessibility tour 'GoVirtually', that is designed to help people with physical and cognitive disabilities. Guidance from Giraud (2015) was used when creating on-site and online information material for **hertz** such as using 14-point size fonts. For events where the artist or collaborators held discussion sessions, a British Sign

Language interpreter could be requested, and subtitles were added to the video made of **hertz** at the Oxford IF Festival described in section 5.

Robson is a keen advocate of physical and interpretative access. Her work at times plays with more than one sensory aspect, this is an aesthetic decision and does not stem from a deliberate intent to make the work more physically accessible per se.

However, **hertz** enhances accessibility for audiences in that one can both feel and hear the processed infrasound.

## 5 hertz tour

The tour of **hertz** occurred at three tour locations: The Oxford Science and Ideas Festival, Tramway Glasgow, and We the Curious at Bristol between October 2018 and February 2019. This section describes the format and setup of **hertz** at each location.


### 5.1 Oxford Science and Ideas Festival

The first tour location was at the Oxford Science and Ideas Festival on 15th October 2018. Given that the emphasis of the festival was on science and ideas it was appropriate here that the set up for the sessions was more educational than artistic. Thus, the scientific research was given more weight than the aesthetic side of **hertz.** The festival organisers allocated spaces

to the diverse events happening during the festival based on their size and technical needs. As can be seen in the video commissioned and produced by Oxford Contemporary Music (https://vimeo.com/306844807) **hertz** was in a room as opposed to a gallery space with the equipment such as Raspberry Pi, and infrasound sensors on show where their functionality could be pointed out for discussion. **hertz** was presented on one day with three bookable sessions throughout the day. During these three bookable sessions, visitors were able to meet the artist and collaborators and interact with the artwork, ask questions,

react, and explore the concepts and research behind the work, and give feedback. The audience was largely made up of families and those with an interest in science, as shown in figure 8. Each session was run so Robson introduced the project and gave background information before handing over to Chaplin and Marlton who explained in detail about their respective research and its connection to the two pieces that make up **hertz**.

**5.2 Tramway Glasgow**

The 2nd stop of the **hertz** tour was Tramway, an art gallery space situated in Glasgow, which ran from the 18th to 21st October 2018. Here a more sensorial, experiential encounter that emphasised **hertz**'s aesthetic and conceptual aspects was pursued. Figure 9 shows an image of the infrasound piece from **hertz** installed in one of Tramway's Gallery spaces. Robson had discussions with Tramway's curator and explored the possible gallery spaces available via video call before deciding on where

the two pieces that make up **hertz** would be installed. Since the install of **hertz** was complex and was to be undertaken by Tramway's technicians without the presence of the collaborators a step by step instructional video was created. In addition to this a troubleshooting flowchart to follow in case of technical issues during the exhibition.

The size of the space chosen for the infrasound piece gave space to each element - the subwoofer and each piece of furniture. At one end of the space was a semi-circular window overlooking an expanse of Glasgow from which the infrasound recorded in real time was being generated from. A bench that had no transducers attached was placed in this window allowing visitors to overlook Glasgow beneath them while they experienced its inaudible symphony from the Subwoofer. The two pieces of furniture with transducers attached were further into the space. The Raspberry Pi and infrasound sensor were hidden from view for safety apart from the leads running across the floor to two pieces of furniture. It was envisaged that the sound from the subwoofer could be heard in the foyer downstairs and would entice people towards the exhibit. On the way into the space visitors entered through a short corridor pass an interpretation board with an overview of **hertz.** The information on the interpretation board was deliberately minimal and had more conceptual than technical information. This was to see how the work presented as solely a work of art. The idea was that the feedback from visitors would then be applied to the much longer exhibition run at We The Curious.

### 5.3 We the Curious, Bristol

The final stop of the tour was at We the Curious, Bristol between November 2018 and February 2019. We The Curious is a science venue with a dedicated space (The Box) for artworks and **hertz** was the venue's first commissioned piece. The box is a small standalone gallery within a science museum, and it was decided that the infrasound piece would work well here. The other part of **hertz**, featuring the Chladni plates, were installed one floor up close to the planetarium to catch people interested in space. As a dedicated recently built gallery space The box has several advantages for installing an artwork and deciding on aesthetics. It was possible to lay the cables to the transducers under the floor, there was also a dedicated lighting rig for the space and the possibility for projection. Figure 10 shows **hertz** setup in the box at We the Curious

Like Tramway the sound of the processed infrasound emitted by the subwoofer could be experienced as visitors entered the building with the sound drawing them towards the exhibition. As they entered the open doorway of the box they saw what looked like three pieces of free standing metal furniture each spot lit, with a further spot light above the speaker on a plinth at roughly chest height. A projector displayed text about **hertz** on to the wall facing them. Otherwise the space was dark but full of undulating sound that immersed visitors in different qualities that shifted depending on your location within the space. Due to the changing infrasound recorded and replayed in real time in the space the sound that filled the room would vary in intensity and tone constantly, sometimes quiet, and gentle, sometimes deep, and loud. The visitors had a choice to sit for a while. Depending on the piece of furniture they chose they would experience strong, light or no vibrations through their body from contact with the metal of the chair or bench in sync with the louder emissions from the subwoofer. They could also approach the subwoofer and feel the gusts of air being displaced in front of it or feel the vibrations by touching it.

Following feedback from the Tramway at Glasgow (section 6.2) scientific information was included alongside conceptual interpretation outside of The Box. In addition to the information projected onto the back wall inside. Postcards with relevant

images, including figure 3 and brief facts about infrasound were available. Invigilators were also briefed with information on the project. The roving educational team was briefed on **hertz** and the project was included in educational demonstrations of exhibits at We The Curious when they happened.

Table 2 summarises the tour dates and the numbers of visitors. At all three locations a table was present where visitors could leave feedback and take away postcards with key facts about the research. To assist in promoting **hertz** to the public and increase awareness of **hertz** at the tour locations **hertz** was promoted through a twitter account and website (https://julietrobson.com/blog/), launched mid-2017. The website also included information about the science behind the project. It was also promoted by each venue through their publicity outlets and by co-commissioners Oxford Contemporary Music and Unlimited, the core project funder, on their website and social media.

## 6 Tour feedback

As discussed in section 5 there was the opportunity for participating members of the public to leave feedback on flip charts or post cards. Whilst there was much audience participation with the piece. Feedback participation varied from venue to venue, like participation, feedback was optional. Here we will briefly describe the kinds of feedback received at each venue and then concatenate all the feedback together to see the overall impression the piece had on the audience.

### 6.1 Oxford IF science festival

The Oxford IF science festival had pre bookable sessions which became fully booked well before the event. This was a good indication that the publicity material was effective. More feedback was received in person than using the feedback materials left out. The video commissioned and described in section 5 documents the day and the reactions of the public. The visitors shown in the video appear to enjoy the **hertz** installation with many of the expressions on their face being of one enjoying themselves and intrigue. During the video one young visitor left a comment on a whiteboard saying, "I wasn't into physics really, until now!". This comment is positive, given that the aims of the project were to build an informative artwork which would also raise the profile of the physical sciences behind the project. One young visitor was so inspired that they got back in touch to do work experience week with the co-authors and spend a day at the University of Reading's Department of Meteorology. The feedback left by the public was generally positive and keywords from their feedback are studied in the thematic analysis in section 6.4.

### 6.2 Tramway, Glasgow

For the period of the tour at the Tramway Glasgow, audience footfall and perceptions were not returned by the venue meaning we cannot report back on these aspects. The only information regarding visitor attendance was reported by the Gallery staff who said that a significant number of visitors purposely entered the Gallery to see **hertz** rather than visit the gallery as a whole.

The following feedback was reported back from Jo Walmsley Tramway's curator and Jo Verrent, senior producer at Unlimited In response to the install of **hertz:**

Walmsley highlighted a few issues; firstly the placement of the main interpretation panel, which was between the works but can be missed by those entering the space from some access routes. In retrospect she would have had two and placed one by each element. The second was an ongoing issue with invigilators who varied in their responses, when visitors wished to discuss the work in more depth not all answers could be provided. Finally, Walmsey reflected that if they were to restage the work again, they would make a separate 'relaxed reading' area that could provide more information for those that wished it.

Verret commented 'I think it's really interesting how much any artist adds to or distracts from their work, and how much they should or shouldn't be present - it's a huge balancing act as audiences all want different things! I personally missed your wonder and excitement that lies behind the work a little so think it's interesting to think how this might be brought in a little!' The idea was brought up between Walmsley and Verrent that a video introduction to **hertz** was a possible solution to this.

In summary the feedback from Tramway was that more Interpretation information regarding the scientific and technical aspects could have been present and that the experience could have been improved by having the collaborators visit for a questions and answers session.

**6.3 We The Curious, Bristol**

The exhibition at We The Curious had by far the largest amount of public engagement this is likely due to the 3-month period it was installed for. We The Curious staff were able to provide some more in depth analysis and were able to provide such statistics as: the average amount of time spent at the infrasound exhibit was 8 minutes and 3 minutes for the Chladni plates, and that the majority of people sat on both pieces of furniture during their visit, broadening their experience. Further to this 95% of people who visited engaged with both the infrasound and the Chladni plates. We The Curious was also able to poll people's opinions and they found that 85% of visitors said they felt a stronger connection to the hidden sounds of the earth after visiting the exhibit, and 91% said they felt they understood more about infrasound based on their interaction with the piece.

**6.4 Overall evaluation of feedback**

In this section we discuss the content of the feedback from all venues and seek to find out firstly if people felt a stronger connection to the hidden vibrations of the planet, and secondly explore the kind of connection this forged with the audience through a thematic analysis to discover underlying themes from the feedback received.

Data from We the Curious showed that 85% of the 6786 visitors felt a stronger connection to the hidden sounds of the Earth. This in short answers our question about whether audience members felt a stronger connection to the hidden vibrations of the

planet. To explore the kinds of connection made we undertake a thematic analysis. Braun & Clarke (2006) describe a thematic analysis as a method to discover patterns and themes in qualitative data, for example interviews with people regarding an experience. Here we perform a thematic analysis on the written feedback received at the venues. First the written feedback was scanned to search for descriptive words. Figure 11 shows a word cloud showing all the descriptive words gathered from the feedback.

Table 3 shows a thematic analysis of the feedback received. The feedback received was categorised into 11 themes as shown in column two of Table 2. Audible intensity and tactile themes in the feedback show that participants are relating to one of the main themes of the **hertz**, vibration. The themes of Calmness and a sense of grounding indicate that participants felt that **hertz** created an environment which made them feel more grounded, potentially due to the awareness of their normally inaudible surroundings. The themes astonishment, thrilling and captivating indicate that the project generated interest in the science behind the project in terms of highlighting that infrasound propagates throughout the atmosphere from many different sources from the aurora to glaciers and is yet inaudible to humans. The themes Frightening and Sense of the Unknown also show that the installation is still making the participants aware of the inaudible world around them and making them aware of the science behind the project. However, the theme frightening suggests that they also have a respect for the delicate yet relentless natural world around us as they experience **hertz**. A final theme, 'Irritable', was found indicating that perhaps **hertz** was not for everyone and that the **hertz** installation caused some participants some discomfort. This could be seen as a barrier to them reconnecting with the hidden vibrations of the Earth.

Due to the logistics of the installation being at We the Curious for three months, and availability for Tramway it was not possible to have Q&As with the co-authors. This leads again to some feedback such as: "I'd love to know more about how you actually interpret that sound. Like, have you just fudged it? Or am I genuinely listening to the sounds of Bristol?" and "I think it's a really nice approach –it would be nice to have something actually explain all the science to me –guess you can't just keep a scientist in a box".

In summary an exit poll at We the Curious suggests a vast majority felt more reconnected with Earth's hidden vibrations after visiting **hertz.** The themes highlighted in the thematic analysis showed that people connected with the vibrations being created and had an increased awareness of the infrasonic sounds generated by the Earth around them. This interest had two schools of thought, one is that of astonishment and awe for our planet. The other was more of a fear and shock at the immense power of our planet which produces infrasound. In addition to this feedback suggested science communication could be improved by having collaborators attend discussion sections at the venues.

## 6 Reviewing the science-art relationship

In this section we will focus on how the art science relationship developed over the duration of the project. At the first meeting between Marlton and Robson, Marlton presented a brief introduction on Infrasound with much explanation of terminology through diagrams. Robson in turn explained her background in contemporary art, an interest in sound and the invisible and inaudible frequencies of the Earth and Cosmos. The discussion then turned to what each party wished to get from a potential collaboration. For Marlton it was to create an outreach activity which used infrasound, highlighting its use in weather prediction and to highlight the archives of infrasound that are recorded constantly across the globe. For Robson it was a potential to further explore her research interests in the natural sciences, inspired by her father Dr Michael Robson a plant physiologist, to develop ways that could be used to make inaudible sound tangible using the latest scientific research. Robson constantly asked questions and Marlton would answer in full in what sounded like a different language, equally Robson would talk about concepts in contemporary art and receive blank looks from Marlton. However, a positive attitude by both led to thinking about simplifying the terminology and different ways to explain concepts. By the end of the first meeting a rough concept idea had been conceptualised of a piece that could connect people to the hidden resonances of our planet. The initial discussion stages have similarities with art-science collaborations such as MacMullen (2005) where initially the artist is finding as much about the scientist's research as possible.

The next stage of the project involved visiting each other's working environments, Robson visited Marlton's research institution and Marlton visited Robson's studio to construct some of the early prototypes and work on the signal processing. It was here that differences in art and science projects differed. Webster (2005) describes that many art science collaborations come about through an artist being hosted at a Research Institution. Here, Robson had independently acquired Research and Development funding from Unlimited, a commissioning organisation who fund artists who have faced access barriers in their careers. The caveat of the grant was that the funding be used to explore proof of concept ideas that need not produce a final working prototype. It is extremely important that artists are able to have time and space to do this, that they're able to follow lines of inquiry and experiment with ideas and concepts without fear of 'failure' or constant pressure to produce a fully-fledged artwork which stifles creativity and ambition. Marlton said this was an appealing prospect coming from a background where funding is often dependent on proof of concept and goal-oriented outcomes. Indeed, there were no guarantees that what we conceptualised would even work and the funders understood this.

It would be easy to assume that the relationship between Marlton and Robson was that Marlton was a provider of technology for a preconceived concept developed by an artist. However, this would be to deny the journey that Marlton and Robson embarked on and the ideas that developed for the demonstrations during the research and development phase and the tour.

At first inspection it would seem that the project followed a generalization outlined in Webster (2005) that the scientist would provide a technology and the artist would work on curating the conceptual development and communicating the project to venues and the public. For **hertz** there are some similarities. Marlton led on the technology development side but was more than a technology provider. For example, suggesting that transducers used for gaming could be used to enhance the physical experience. In addition to this Marlton spent time testing the effects of different infrasound filtering parameters and partaking in conversations about how different configurations may alter authenticity of the final processed output. Thus, taking a more creative role. Robson having little to no experience of the technology involved in the sensing of infrasound learnt from Marlton what he thought was and wasn't possible with the technological aspects, this in term informed the development of concepts and what may be possible in the final tour version of **hertz**.

Working on the project over 2 years enabled Robson and Marlton to hone their methods of communicating complex ideas with each other. Understanding of each other's subject areas developed gradually by being around each other and picking things up, an osmosis of knowledge aided with the use of diagrams and developing metaphors for concepts. Towards the end of the project Robson had a deeper understanding of infrasound, its uses, and potential not only for future artworks but for research and its current applications. Marlton had a greater understanding and appreciation for contemporary art, particularly interactive art, and its potential to make complex scientific ideas not only understandable but exciting, relevant, and meaningful.

Jeffreys (2018) states, 'When it comes to collaboration, there is a danger that generalizing about scientists as: rational, institutionalized and ends-oriented versus artists: emotional, free, process-driven can risk accentuating disciplinary stereotypes. Or, conversely, that trying too hard to find commonalities can lead to simplistic platitudes about 'creativity'. People are more than the discipline they represent.' During this project such a generalisation as this was avoided and we felt the metaphorical wall that exists between the two disciplines was overcome, similar to case studies discussed in Leach (2005).

It became clear that certain methodologies were recognisable to both Robson and Marlton. Both worked with flexible forms of praxis-based research which could be defined as a form of critical thinking and comprises the combination of reflection and action taking. This reflection and action taking is demonstrated in section 2.3 when it came to assessing how to best process the infrasound from feedback during the prototyping sessions. Ultimately the project had similarities to that concluded by Stewarts (2003) who stated, "The research function of developing and extending knowledge is to be judged on the products of that research. In the same way that a learned paper is evidence and coherent argument for all the processes that preceded it, laboratory or speculative, the finished work of art is the culmination of the theory and practice of the discipline. Based essentially on investigative, exploratory, speculative, or analytical processes, the outcomes are a result of synthesising the problematics of the discipline. Like the best research in any field, it is expected that creative work will comply with the defining characteristics."

It was previously highlighted that at the start of the project that Marlton and Robson, whilst being positive, were unfamiliar with each other's subject areas. We reflect here on how these attitudes have changed. In an interview with Liz Hingly for the University of Birmingham's Phyart website Robson stated:

*"The project (hertz) is the first time I have worked in depth with scientists. Overall, it has given me an understanding and deeper appreciation for the ways artists and scientists can collaborate, affect each other's work, and learn each other's language. This can only be achieved through a commitment by the artist to meaningfully engage in scientific research and for the scientist to trust in the artist's ability to create something that communicates on both an artistic and scientific level. Finding*
*a successful working partnership, where both parties meet on a common ground is not easy. Simultaneously the partial mystery in each other's processes of research and creation motivates the relationship. ... My main motivations are to make an outstanding artwork with the use of cutting-edge science that is accessible to as many people as possible. Something interactive, intangible, mysterious, an encounter that inspires questions. I don't aspire to be a physicist; I want to retain and share the mystery of true science."*

Robson states a deeper appreciation of science art collaborations due to the work on **hertz** and the statement above reflects strongly on the creative development processes discussed here. It also shows how Robson has succeeded in her quest to bring an artwork using cutting edge science, Infrasound, to the public. Marlton was also interviewed by Robson as part of the project about how the project has changed his attitudes:

*"If I told you I had made a chair that vibrates to the sound of the earth, people might have been, like, so what? But with the use of Juliet's (Robson) experience, hertz has been presented, raised in profile and her creative license and skills have been used to create an installation that engages and enthrals the public, I couldn't have done that myself."*

This echoes Marlton's desire at the beginning of the project to create an installation that highlights the use of infrasound that the public can engage in. It also highlights the importance of how working with an artist can be used to raise the profile of research as discussed in Webster (2005). When Marlton was asked about how **hertz** had changed him as a scientist he replied: *"I'm more interested now in different ways of displaying data. I make a data plot and sometimes, I think we could turn it into something you could feel, or hear and it has opened up new ways of visualising data, especially if it is something quite complex.*
*So, I am more open to when you look at data or something thinking what could I do with this? I guess learning from Julie, who looks at something and thinks of several quite out the box things that could be done with it. So, I guess as a result of hertz I now think a lot more out of the box."*

This is an interesting concept and has been discussed in Eldred (2016) and Segarro (2018) who highlighted how scientists who
either view art, create their own art, or work in collaboration with other artists can help boost their problem solving abilities.

This is due to being able to switch focus completely from their own work or view other creative efforts as new perspectives to solve problems.

In summary **hertz** did not suffer from the generalisation that the scientist is a technology provider and the artist is the curator of the given technology. Here that boundary did not exist, and the scientist had a large input into the creative part of the project. Likewise, the artist took an extensive interest in understanding the science involved and was able to input into the design of the technology developed for the project. This was achieved by having a positive attitude and an open mindedness to learn about completely new subject areas. When collaborating a similar methodology of action taking and reflection provided a framework for creativity. In addition to this, having a grant that was not issued by a research institution as is commonly done allowed the development of **hertz** to take a more dynamic route where creativity was prioritised over the need to produce a finalised artwork by a deadline. For Robson **hertz** allowed her to gain experience in art science collaboration that can be applied to future art-science collaborations. For Marlton, his knowledge of contemporary art has increased, along with seeing the benefit of working with artists in the future to boost the visibility of research. It has also opened new avenues in problem solving.

## 8. Conclusions

Here an art installation, named **hertz** with an aim to allow people to feel reconnected with the environment and be used to help describe the science of infrasound was created. This was achieved through a successful research and development phase that allowed ideas to be developed. The subsequent three stop tour of the UK allowed **hertz** to be exhibited in an accessible way to the public. Analysis of the feedback received showed that most participants found a deeper connection to Earth and the environment after partaking in **hertz**. For a future tour of **hertz** or a variation of, an installation could be set up in an underground station where large sections of the space vibrate and shake. **hertz** also demonstrates that there is considerable potential for outdoor structures and street furniture to have transducers attached to, widening opportunities for experiential art. In addition to this more thought is needed into how to communicate the science behind the installation whilst not detracting from its aesthetics.

The collaboration between the artist and scientist worked well. The key to a successful collaboration was to keep a positive outlook and break the generalisations of scientists being a technology provider and the artist the curator of the technology. Here both artist and scientist broke the metaphorical wall using common methodologies between the two subject areas which led to a more creative output. The experience of the collaboration has improved each individual's working practice. For the artist it allows more confidence in forming new art-science collaborations for future works. For the scientist it allows new perspectives for problem solving. The experience documented here can be used as a model for future art-science collaborations.

The authors encourage more Science Technology Engineering Art and Maths (STEAM) projects (Seggara et al. 2018) based on their experiences here.

## Acknowledgements

R&D Grant from the arts commissioning body Unlimited (https://weareunlimited.org.uk). Co-commissioned and supported by Unlimited, celebrating the work of disabled artists, funded by Arts Council England. Co-commissioned and supported by We The Curious and Oxford Contemporary Music. Supported by The University of Birmingham. The Friends of the University of Reading provided a small grant to fund the furniture and hardware used during the project. During the project GJM was funded by the ARISE2 project, a collaborative infrastructure design study project funded by the European Commission H2020

program (grant number 653980,arise-project.eu). Special thanks go to Giles Harrison, for his constructive input into helping shape this manuscript and sharing his experiences of art-science collaborations. Martin for his support during the 101 outdoor arts residency and for carrying out the onsite installs and strip downs. Kate Stoddart in her capacity as producer of **hertz** R&D and expertise in strategic planning for the tour and independent producer Bill Gee for his advice and support. The **hertz** tour was part of Season for Change 2018, a UK wide programme of cultural responses celebrating the environment and

inspiring urgent action on climate change in the lead up to the UN Climate of Parties 'COP24' talks taking place in December 2018, which were critical in meeting the targets of the Paris Agreement.

## Ethical considerations

All feedback received from members of the public had been anonymised where needed. At events where photos and videos were taken an opt-out policy was adopted where people who did not want to be in the photos or videos were asked to make

themselves known or wear a wristband so the photographer knew not to include them in frame. People experiencing the artwork chose to do so at their own will and were free to leave or stop partaking at any point.

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

**Figures**

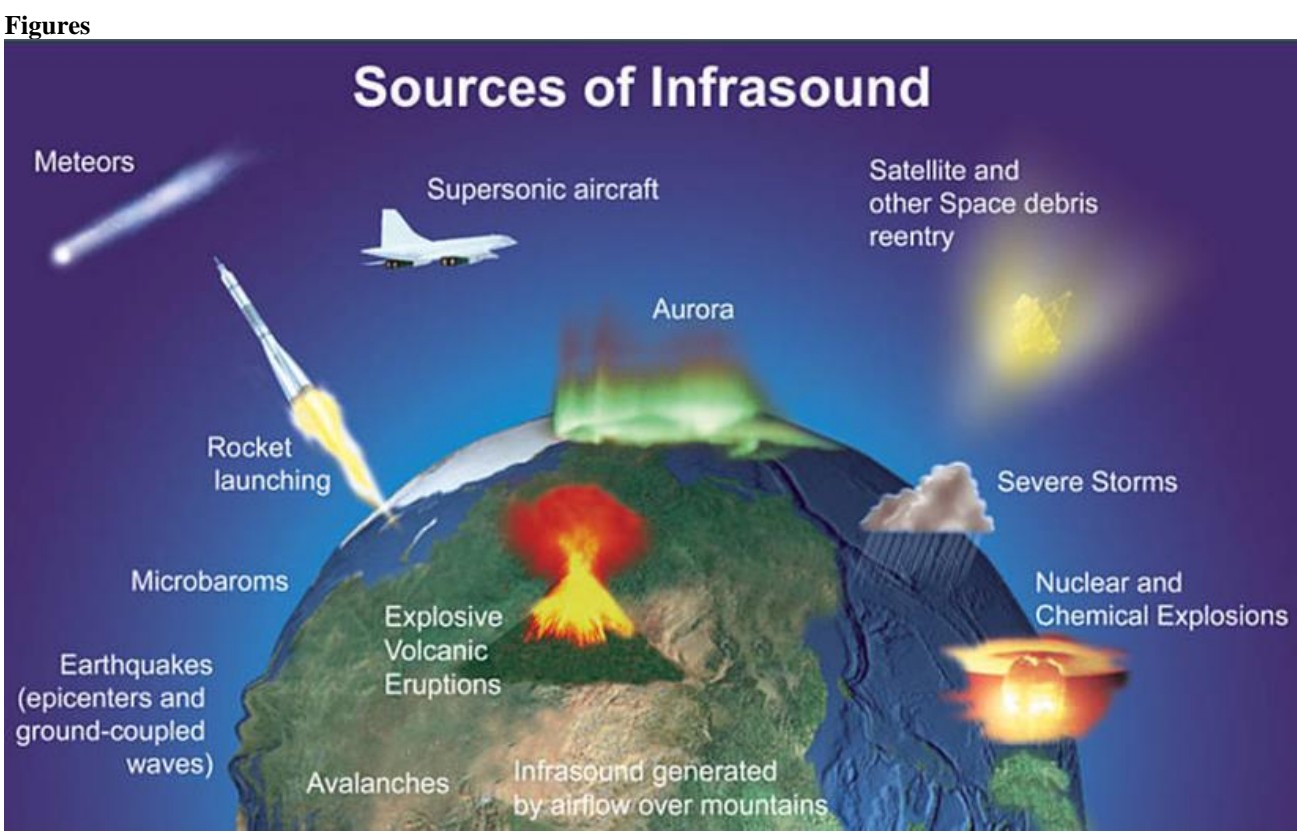

**Figure 1: Sources of infrasound both manmade and naturally occurring from ARISE website (arise-project.eu)**

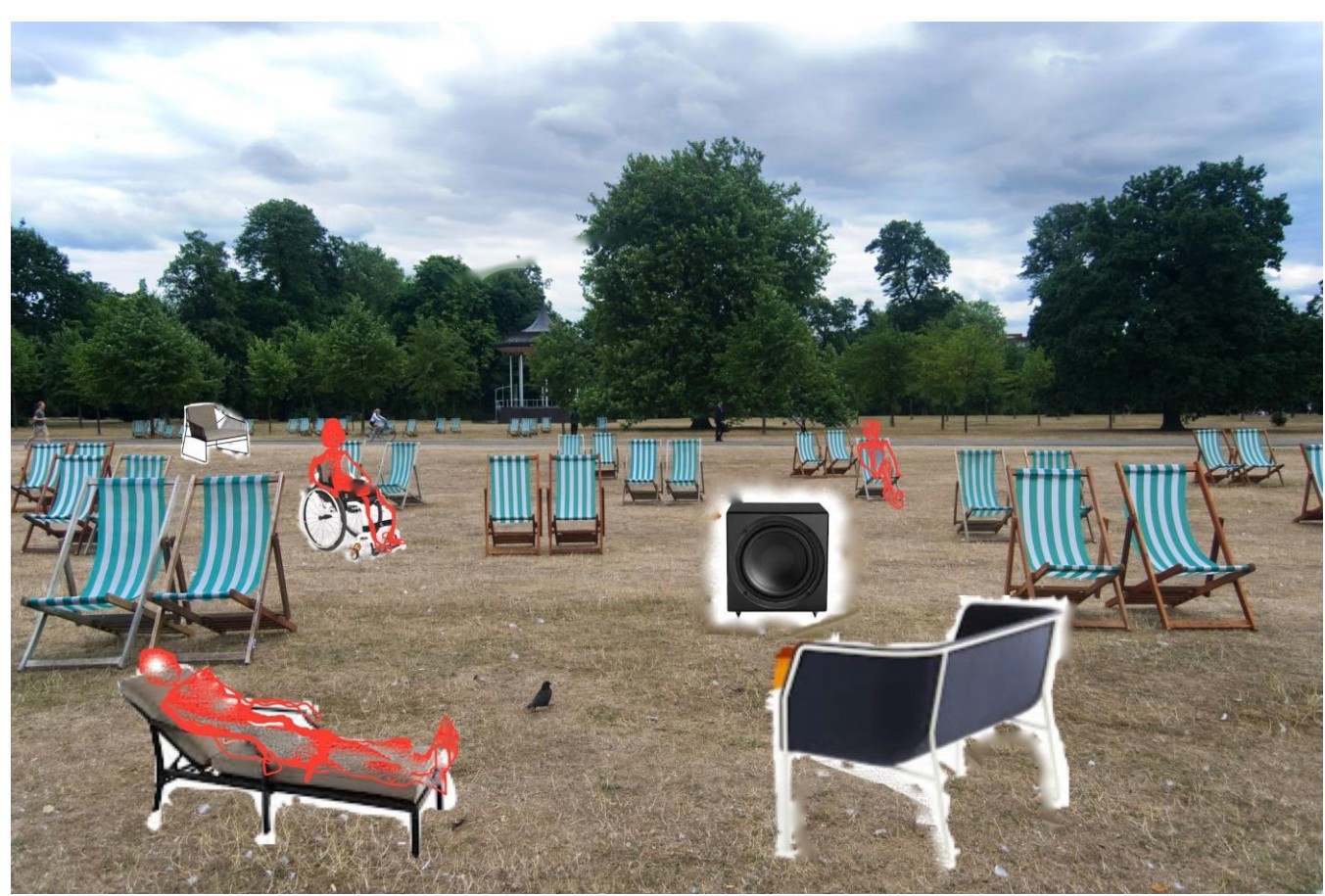

**Figure 2: One of Robson's initial drawings of hertz in an urban park. Blue and white striped deckchairs, other furniture, and wheelchair; three people sitting, one in a deckchair, one using a wheelchair and one lying on a sun lounger. Figures are drawn in red. Among the furniture is a large subwoofer. The furniture would have transducers attached to vibrate them and the subwoofer would play the sounds of a large ocean wave or storm a long way off the coast.**


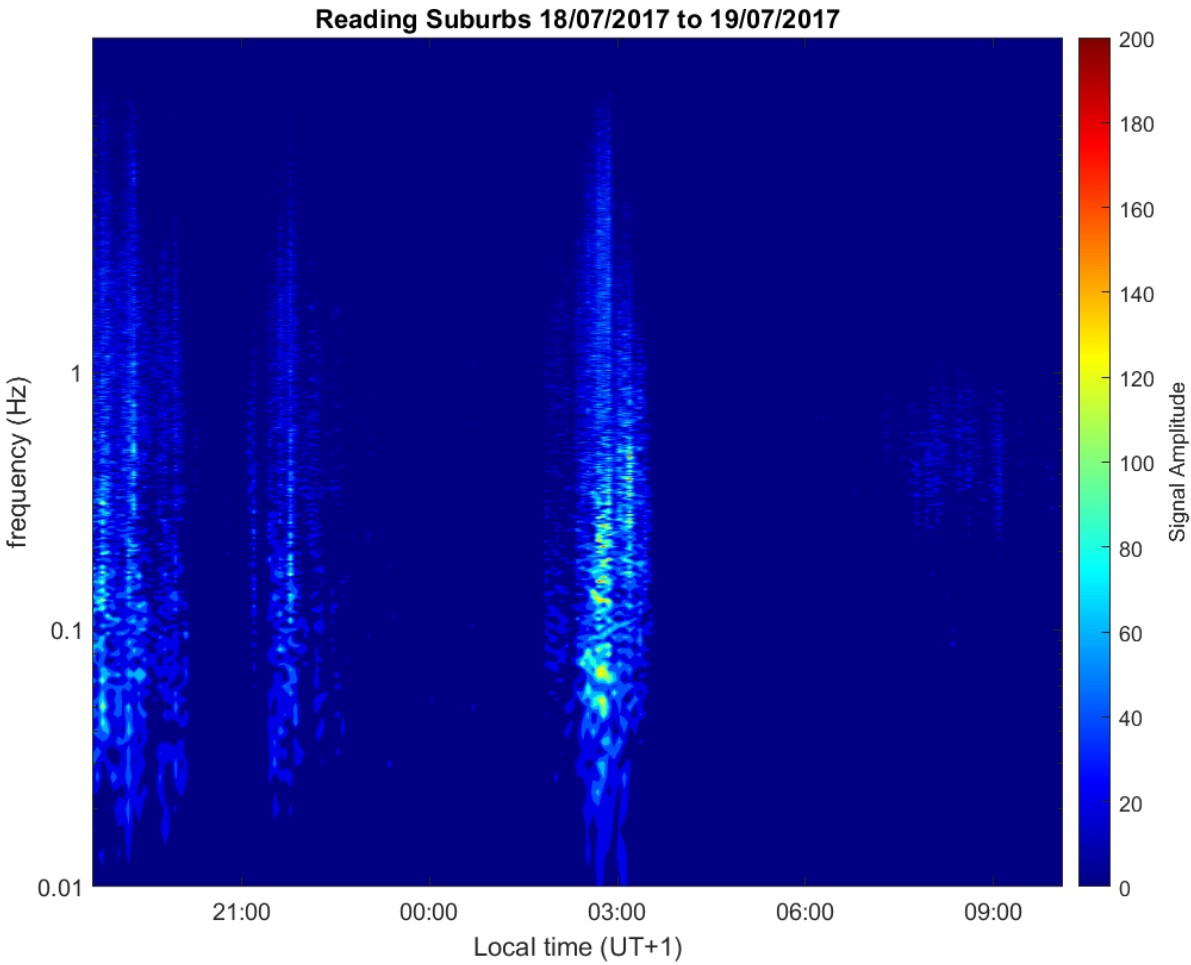

**Figure 3: A spectrogram of infrasound data obtained by an INFRA200 sensor placed situated in the SE Reading suburbs from 18th July 2017 to the morning of the 19th July 2017. (On the x-axis is local time in hours on the y-axis is the frequency of the infrasound signals. The colour bar on the right shows the strength of the infrasound signals.)**


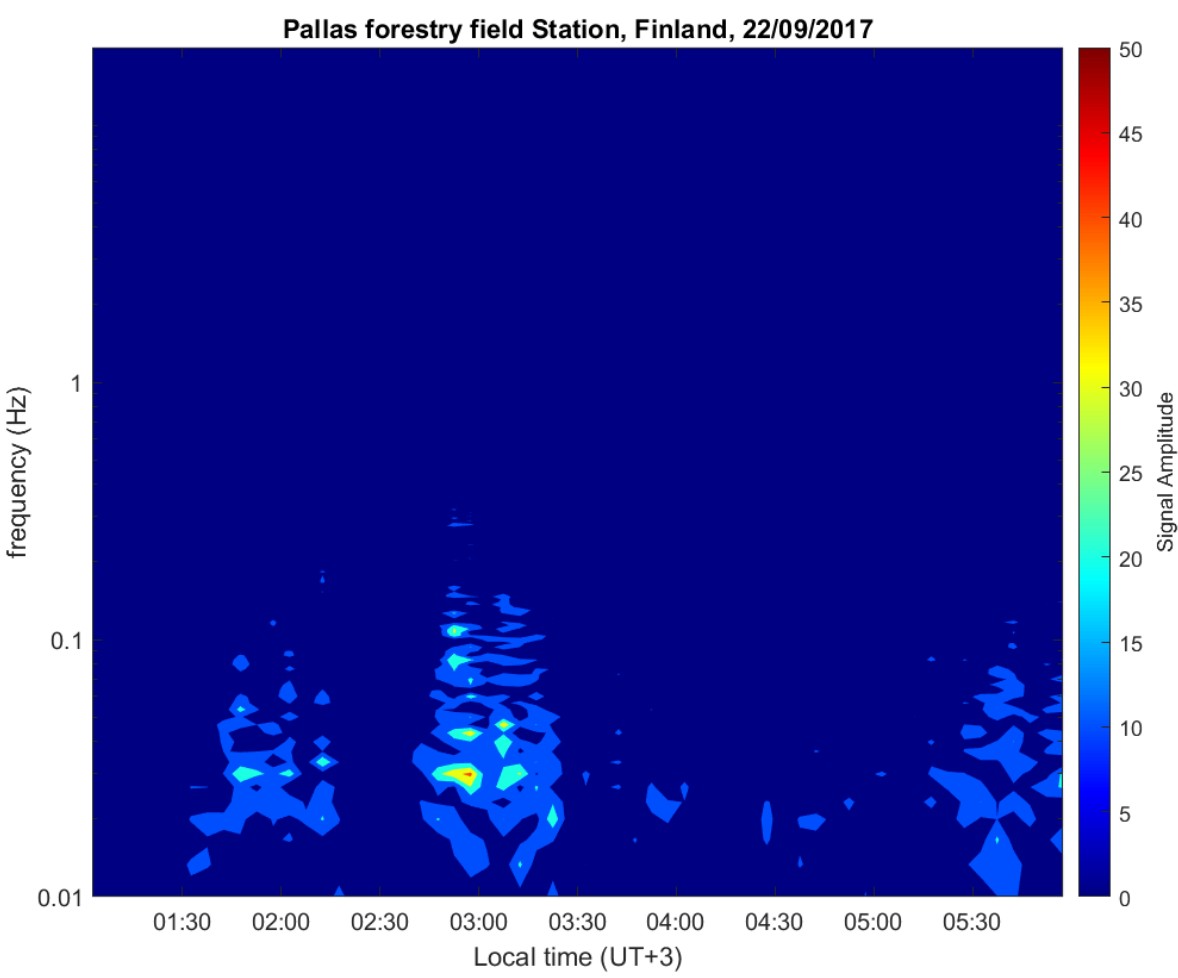

**Figure 4: A spectrogram of infrasound data from the infrasound sensor placed situated in Pallas Finland on 22nd September 2019. (On the x-axis is local time in hours on the y-axis is the frequency of the infrasound signals. The colour bar on the right shows the strength of the infrasound signals.)**


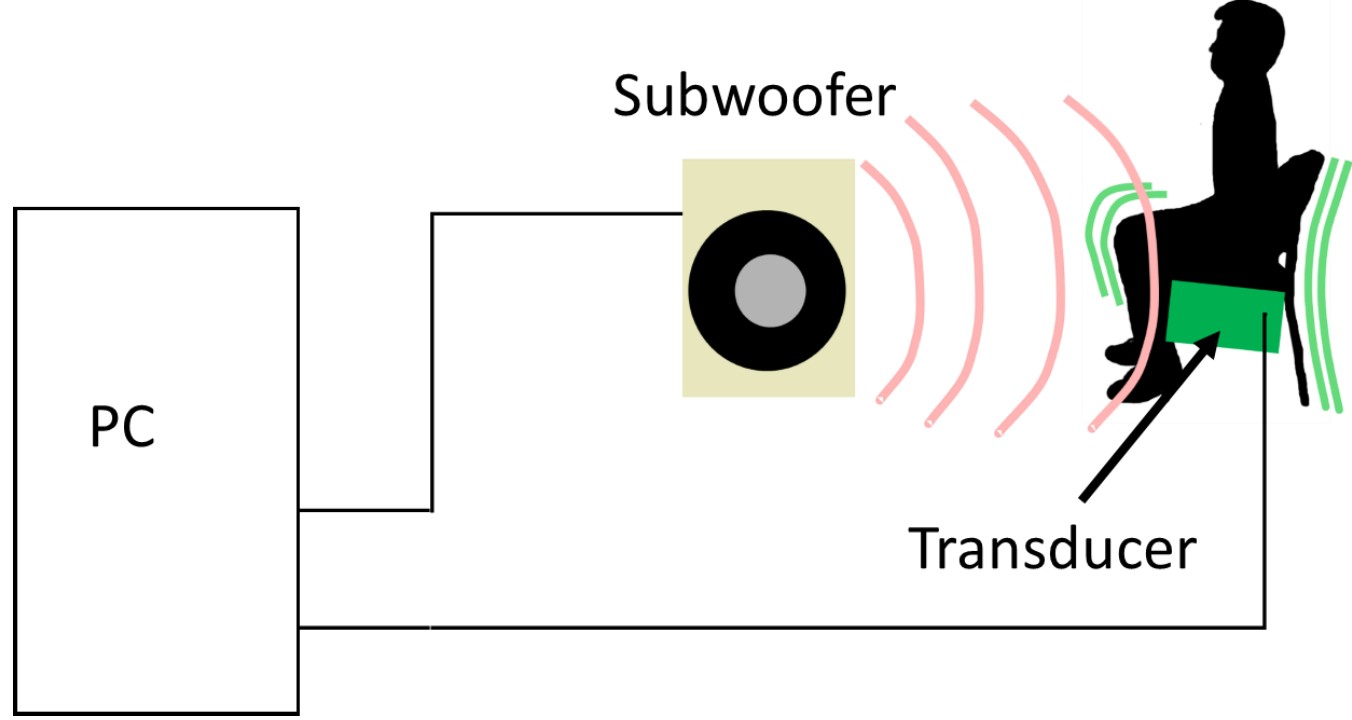

**Figure 5: Schematic of hertz prototype rig.**

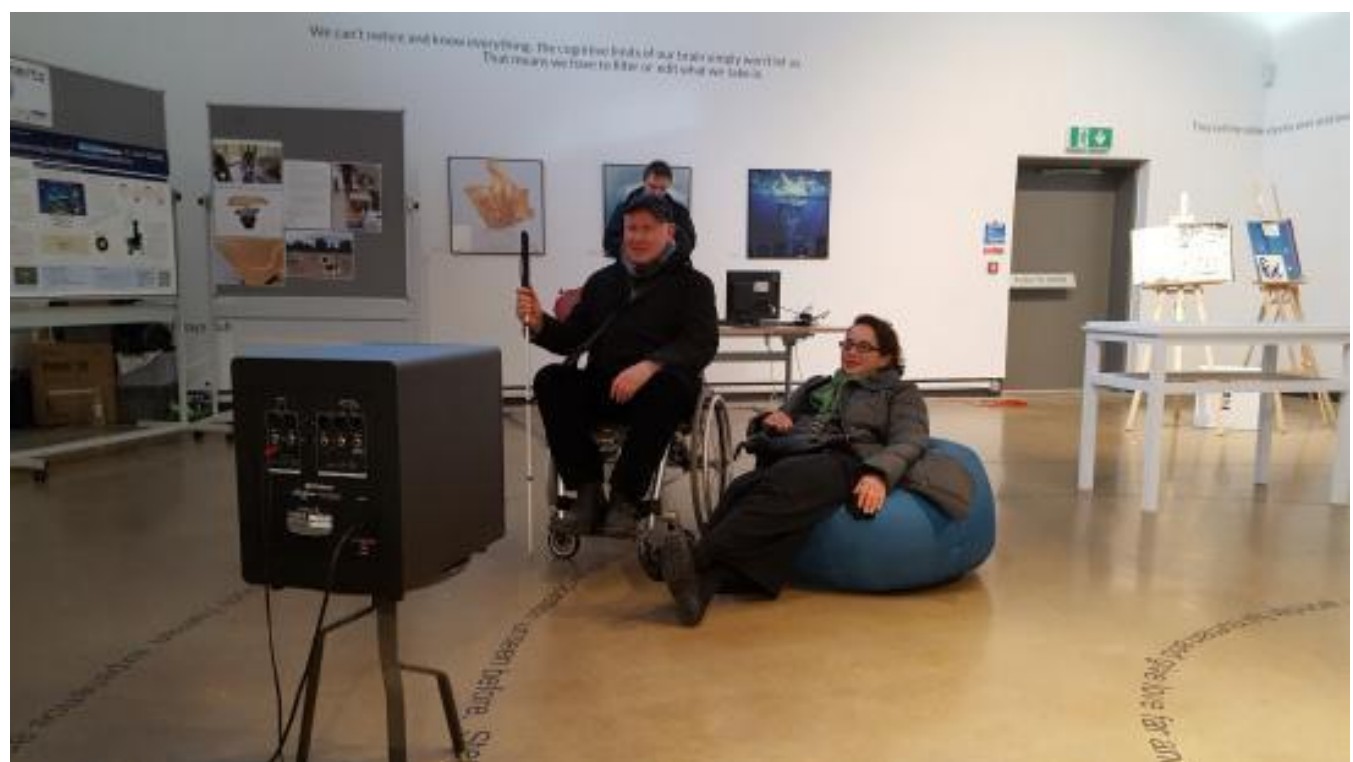

**Figure 6: Two people experience the effect of infrasound played through the transducer and large subwoofer in a gallery space. They both face a subwoofer that fills the room with low frequency reverberation of sound.**

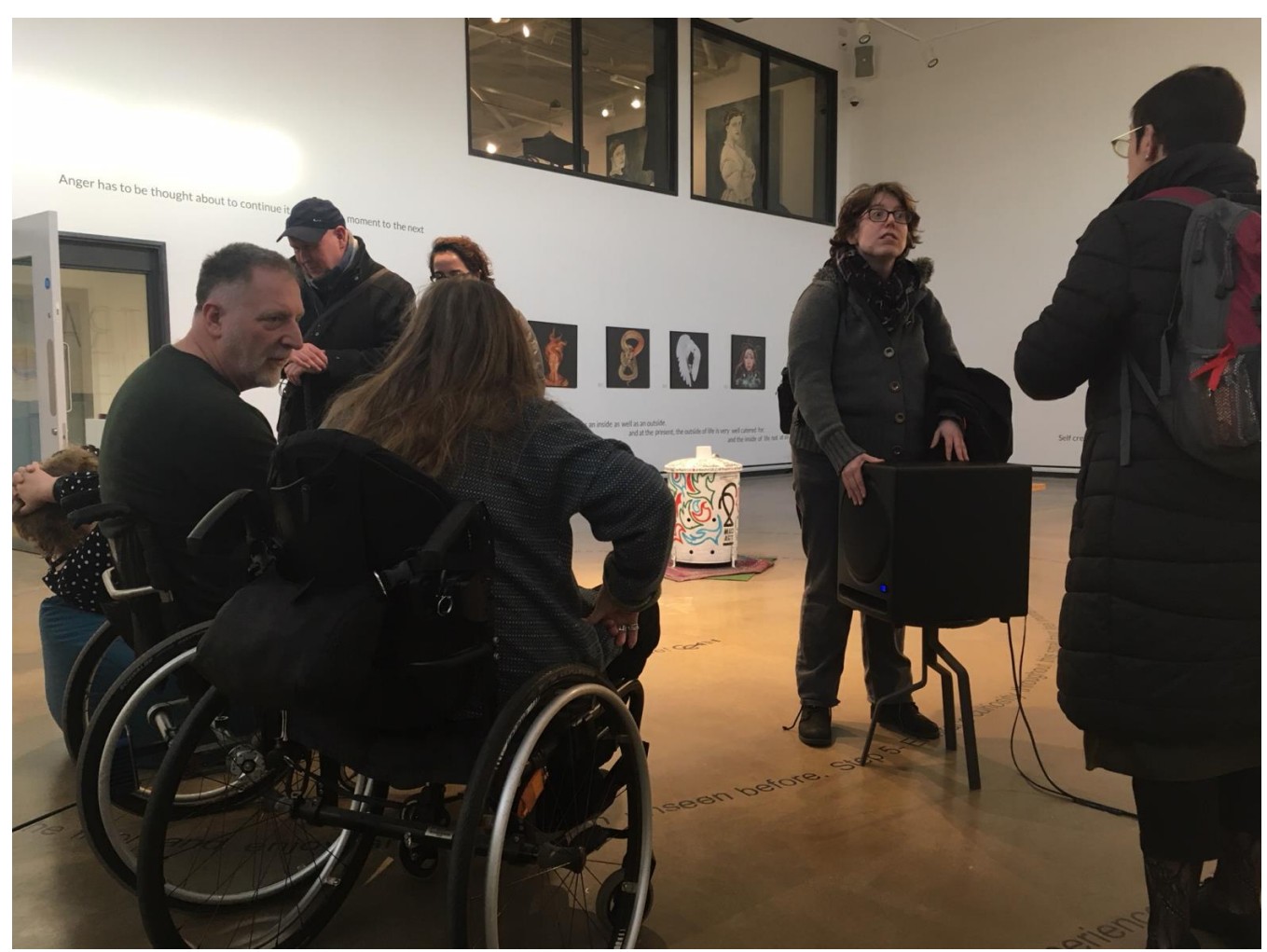

**Figure 7: A participant who is deaf places their hands on the subwoofer to experience the processed infrasound at be there at the start festival, Attenborough Centre, Leicester, credit Glenn Bryant**

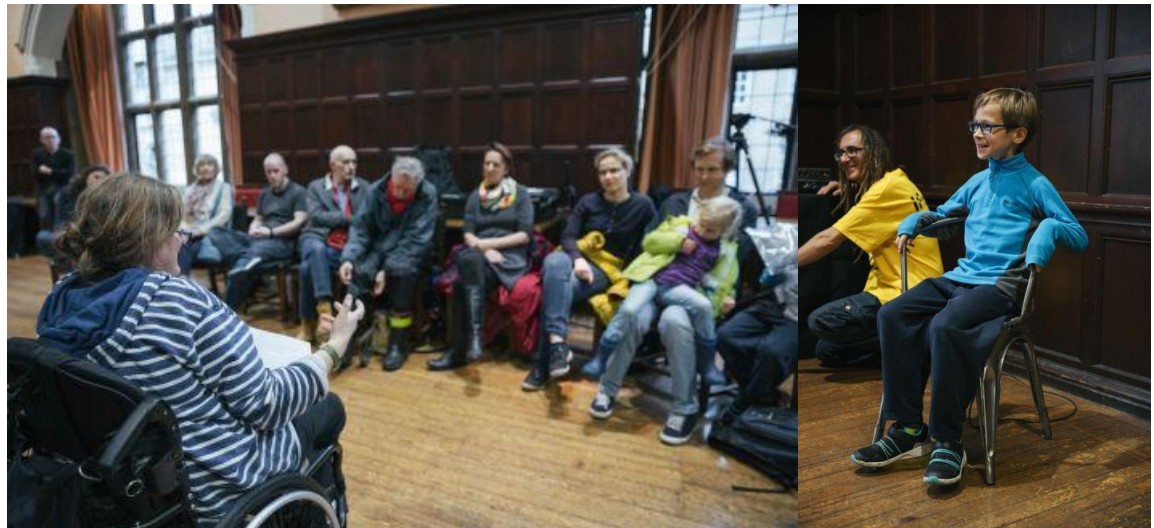

**Figure 8: Left: Juliet Robson introduces hertz to the public at the Oxford Science and Ideas Festival, October 2018. Right: a young member of the audience experiences the reverberations of infrasound from the Oxford locality through one of the metal chairs.**


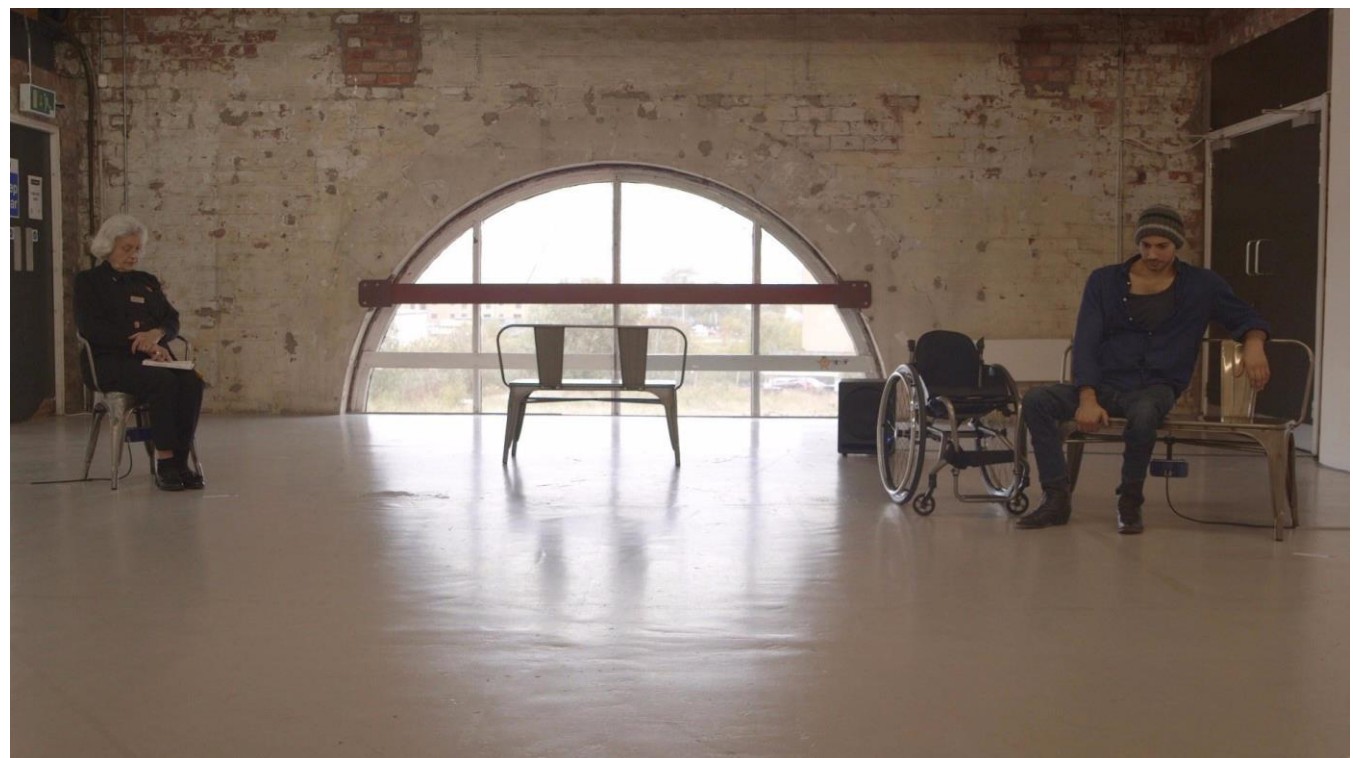

**Figure 9: Finalised version of the hertz infrasound installation at the Tramway, Glasgow. Two members of the public sit on the furniture which has the transducers (blue) clamped underneath. The powerful subwoofer is in the background.**

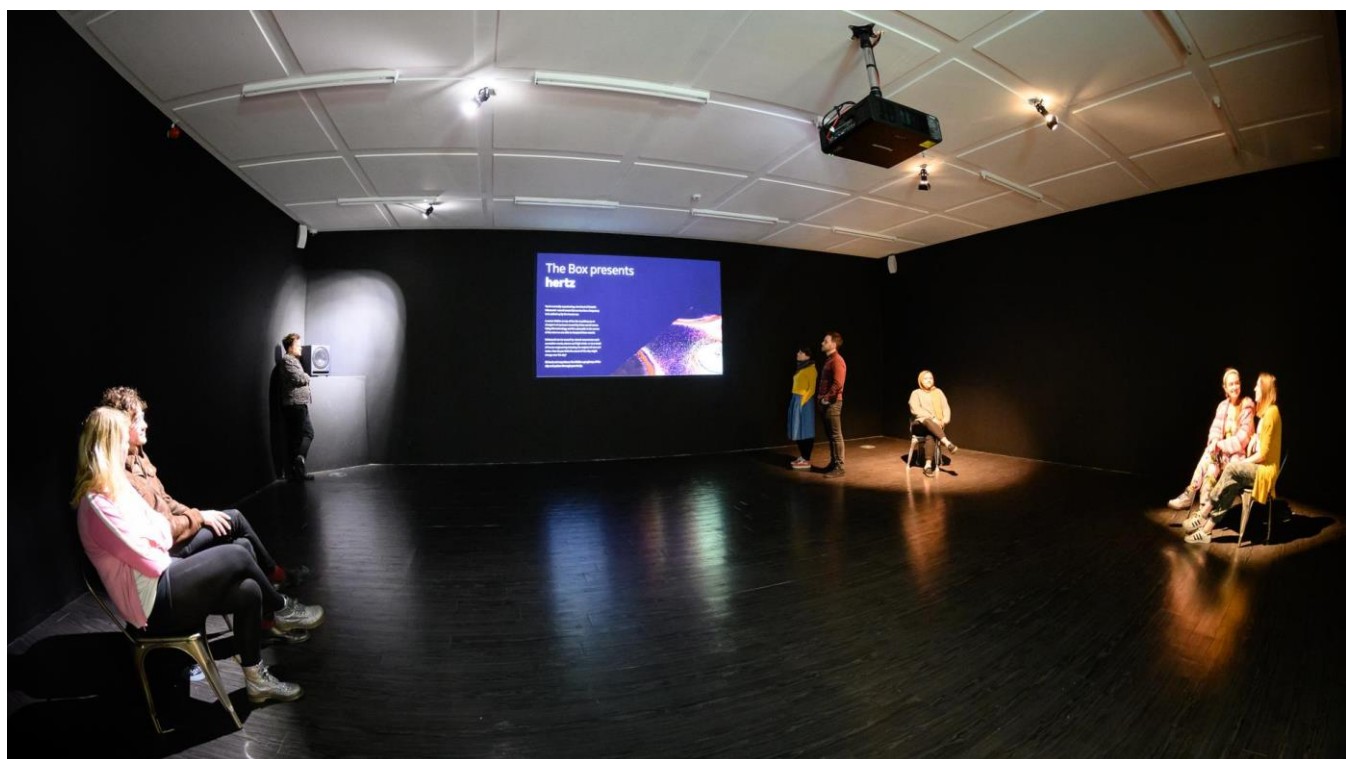

**Figure 10: Finalised version of the hertz infrasound installation at the box at We the Curious Bristol. Members of the public sit on the furniture which has transducers beneath. The subwoofer is located in the far-left hand corner. Information about hertz is projected on to the wall**

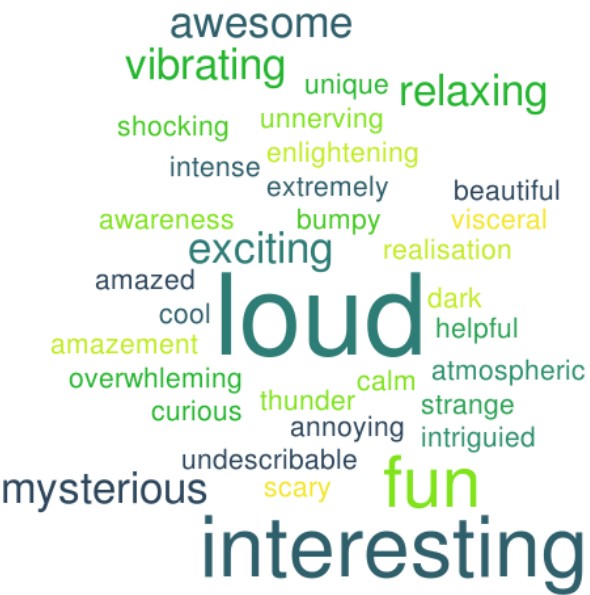


**Figure 11: A word cloud summarising the words used to describe the hertz project by the public during its tour.**




**Tables**

| Date | Location | People | Audience |
|---|---|---|---|
| November 2017 | Wyfold Lane studio, Oxfordshire, UK | 11 | Funders of R&D, potential hertz supporters and programmers and Scientists. |
| March 2018 | Wyfold Lane studio, Oxfordshire, UK | 25 | Local families and school children |
| March 2018 | 101 outdoor Arts, Newbury, Berkshire, UK | 15 | Resident artists, Art Commissioners and bbc 3 presenter Max Ernst |
| March 2018 | Be there at the Start Conference, Attenborough Arts Centre, Leicester, UK | 100 | General Public |
| April 2018 | Session EOS8 – Scientists, artists and the Earth: co-operating for a better planet sustainability, EGU 2018, Vienna, Austria | 35 | Scientists |
| April 2018 | We the Curious - After Hours Event, Bristol UK | 50-70 | Artists |

**Table 1: List of public engagement activities where the hertz prototype were shown**



| Date | Location | Audience Numbers |
|------|----------|------------------|
| 15/10/18 | Oxford Science and Ideas Festival, Oxford, United Kingdom | 55 |
| 18/10/18-21/10/18 | Tramway art-space, Glasgow, United Kingdom | Unknown |
| 7/11/18-28/3/1 | We the Curious, Bristol, United Kingdom | 6786 |

**Table 2: Hertz tour locations and audience numbers**

| Keyword | Theme |
|---------|-------|
| **Loud, Extremely loud, Intense, Overwhelming, Thunder** | **Audible intensity** |
| **Bumpy Vibrating** | **Vibration** |
| **Realisation, Enlightenment, awareness, visceral** | **Sense of grounding** |
| **Relaxing, Calm, Atmospheric** | **Calmness** |
| **Amazement, Amazed, Awesome, Beautiful** | **Astonishment** |
| **Fun, Excitement, Cool** | **Thrilling** |
| **Scary, Shocking, unnerving, dark** | **Frightening** |
| **Unique, Indescribable, Strange, Mysterious** | **Sense of the Unknown** |
| **Intrigued, Interesting, curious, engaging, interactive, Fascinating** | **Captivating** |
| **Annoying, not fun** | **Irritable** |

**Table 3: Shows a thematic analysis of the feedback from written feedback from the hertz tour.**