# Peer review of "Developing the hertz art-science project to allow inaudible sounds of the Earth and Cosmos to be experienced"

_Geoscience Communication, 2020_

## Referee Comment (RC1) · Anonymous Referee #1 · 6 Apr 2020

The manuscript concerns a project which enabled infrasounds from various sources on Earth to be experiences through a multisensory artistic exhibit. The article focuses primarily on the development process of the exhibit, a collaboration between an artist and scientists. This is relevant to the journal Geoscience Communication and would be of interest to those thinking of similar projects or approaching art-science collaborations. I do, however, have a number of concerns over the information presented which I detail here.

Main issues:

The introduction could do with much more of the broader context of science commu-

nication and public engagement that concerns this area of science or uses a similar method in order to properly frame this project. At present the motivations that people need to re-establish links with the natural environment come across as merely the opinions of the authors and not backed up by any published research or public dialogues. Only with this wider context is it possible to better consider the successes of this project.

The main contribution that this articles makes to the literature is arguably the development process of the exhibit. I applaud the authors for writing this in accessible way, however, interested technical readers may want more detailed information. I suggest the authors provide this in an appendix, e.g. giving precise parameters used in their processing so that others may be enabled to convert similar infrasound datasets.

The evaluation data and its presentation in section 4 are rather lacking unfortunately. There is little to no detail of how "feedback" was collected, what specific questions were asked of participants, and how the qualitative data has been analysed. To this latter point the authors seem to have simply classified whether or not it was positive and provide, seemingly cherry-picked, example quotes. This work calls out for a thematic analysis to better understand what participants' responses to this experience were, what common themes emerged and how do they relate to the aims of the project and compare with other similar efforts? Can any conclusion be made linking back to the aims of the project, e.g. did it reconnect participants back with the Earth?

While the review of the collaboration is also interesting, more discussion and conclusions need to be drawn from the quotes provided.

Specific comments:

Throughout the term "resonance" seems to be used slightly carelessly. It is not clear to this reviewer whether it is truly resonances which lead to many of the infrasounds considered (indeed many of them seem to be rather broadband rather than peaking at well-definied frequencies), nor is it clear whether the transducers' vibrations are causing waves which are resonating within the human body. I would suggest the authors consider carefully each usage of this word and only include it where appropriate (e.g. its usage in describing asteroseismology is correct) and provide references, otherwise other terms such as sound, vibrations etc. should be used.

Line 98: Arguably the enhanced infrasound power goes to a much lower frequency than 0.1Hz in Figure 2, approximately 0.02Hz.

Line 99: LT as Local Time needs to be introduced in the text.

Line 105: There is no visible power enhancement at 1Hz in Figure 4, instead the biggest peaks appears to be around 0.03Hz.

Line 106: This sentence is confusing. You need to specify what quantity you are referring to exactly and whether you are comparing the two events to one another of the reference in the previous sentence.

Line 123: "documented by smart phone" comes across as though the authors made notes using a smart phone, whereas I understand from later sentences they used an audio recording app on the phone. This should be made clearer.

Line 137: It is not clear how the amplitude was measured and used to modulate tones. The authors may want to keep such technical detail to the suggested appendix though.

Lines 143-160: Polyphonic seems to be the wrong term here, since this is defined as "a type of musical texture consisting of two or more simultaneous lines of independent melody" whereas the authors describe modulated pink noise which is not musical or melodious. What the authors describe is surely more of a cacophony than symphony. It would be very helpful to provide sound clips of the different processed versions of the infrasound for the readers to be able to interpret. Furthermore on this point, the authors' descriptions of the sounds come across as a little hyperbolic and would benefit from some other viewpoints.

Line 167: "practicalities of access needs" are raised but no description or discussion

of what these were are given.

Line 179-180: "further positive media coverage" is mentioned but no quotes or analysis of the material are presented.

Lines 247-253: It needs to be stated how all of these were measured.

Lines 271-26: The numbers quoted here are rather meaningless without benchmarking against similar efforts. Furthermore, qualitative analysis of any tweets about the exhibit (not merely retweets or likes) could provide insight into audiences' responses, which is currently lacking.

Line 308: This should say Figure 1.

Line 316: It is not clear who did the interviewing.

---

## Referee Comment (RC2) · Charlie Hooker (Referee) · 8 Apr 2020

Journal: GC Title: Developing the hertz art-science project to allow inaudible sounds of the Earth and Cosmos to be experienced Author(s): Graeme J. Marlton and Juliet Robson MS No.: gc-2020-9 MS Type: Research article Special Issue: Five years of Earth sciences and art at the EGU (2015–2019)

In essence, this article aims to record the creative collaborative process between and artist and a scientist by documenting the research development of a project and its dissemination, with a particular focus on public engagement and a lay-person's interpretation of a potentially awe-inspiring science-based art installation. The central pivot

to the project is the use of infrasound to encourage individuals from diverse age groups and backgrounds to consider the continual 'invisible' movement and vibrations generated by natural and man-made activity within our planet – to reveal the imperceptible. This ambitious idea for an interactive installation is detailed by the authors as follows: Drawing on the premise that everything vibrates, from the smallest atom to the furthest star, their frequencies surround us and yet leave no imprint, hertz would enable people to feel their bodies resonating to the inaudible symphony of our own planet and experience the stars singing and see their sound made visible. hertz's ultimate goal would aim to reconnect us to our planet and place in the cosmos. (68-71) From an artistic perspective, the conceptual structure underpinning the gallery installations created through the project is rooted in ideas of 'the uncanny' and 'the sublime', postulated by philosophers such as Edmund Burke and Immanuel Kant, and demonstrated by artists such as Walter de Maria, Bruce Nauman, Cornelia Parker and James Turrell. From a scientific perspective, the project is clearly aimed at furthering ways of achieving public engagement and refining research already begun using STEM expertise and the ARISE project, based largely at the Meteorological Department of the University of Reading.

Although the article contains some minor grammatical typos throughout (no full-stop, end of 72; were/was, 113; capital To, 150; comma after 1st word, 182 etc. . . .therefore needs full proof reading throughout) it is an interesting account of an innovative project and gives good information regarding its public presentation and outreach feedback. However, although the ongoing collaborative process is, in general, well documented, there is an implication within the article that the scientists are often problem-solving the artist's practical needs, with no in-depth interrogation and analysis by the team of the visual, audible and physical aesthetic of the objects and installations generated by the overall process and how this informs the scientists' own research and insights. It would be interesting, for instance, to have (280-344) much more detail about how the collaborative process altered each member of the team's initial ideas and approaches to his/her own subject. I believe that the article would also benefit from a more in-depth

description of how and why these particular individuals from these specific disciplines began working collaboratively in the first place – what their original expectations were with regard to research - and how they intend to incorporate aspects of the public feedback they gathered to develop this extremely interesting project further. This could be more fully developed in Section 6, where common methodologies (347) would benefit from being described in much more depth, to reveal the successes, failures and critical analysis of each discipline's methodologies and how the combined methodologies systematically achieved the final outcomes and, potentially, a new methodology.

The article explores an intriguing topic, but could give a more rigorous record of the positive and negative surprises generated when two disciplines come together. The art/science project itself seems to offer the public a potentially poetic experience and to be physically engaging. However, from the article, I do not quite understand the gallery context of the immersive audience participation. If I walked into the gallery, exactly what would lead me to sit on the chair and how would I understand the implications of the uncanny and mysterious source that I was listening to? Is it a feature of the work that there should always be information sheets or 'exhibit demonstrators' available for the public, or does the installation reveal its meaning in another more subtle way - more akin to, say, a Joseph Beuys installation? The article would therefore benefit from a passage describing the team's views regarding public engagement methodology – the pros and cons of installing an object which emanates a scientific principle through its construction and physical location without the principle needing to be contextualised by an additional means - as this does not appear to be fully documented or analysed.

Professor Charlie Hooker.
* * *
Journal: GC
Title: Developing the hertz art-science project to allow inaudible sounds of the Earth and Cosmos to be experienced
Author(s): Graeme J. Marlton and Juliet Robson
MS No.: gc-2020-9
MS Type: Research article
Special Issue: Five years of Earth sciences and art at the EGU (2015–2019)

In essence, this article aims to record the creative collaborative process between and artist and a scientist by documenting the research development of a project and its dissemination, with a particular focus on public engagement and a lay-person`s interpretation of a potentially awe-inspiring science-based art installation.

The central pivot to the project is the use of **infrasound** to encourage individuals from diverse age groups and backgrounds to consider the continual 'invisible' movement and vibrations generated by natural and man-made activity within our planet – to reveal the imperceptible. This ambitious idea for an interactive installation is detailed by the authors as follows: *Drawing on the premise that everything vibrates, from the smallest atom to the furthest star, their frequencies surround us and yet leave no imprint, hertz would enable people to feel their bodies resonating to the inaudible symphony of our own planet and experience the stars singing and see their sound made visible. hertz's ultimate goal would aim to reconnect us to our planet and place in the cosmos.* (68-71)

From an artistic perspective, the conceptual structure underpinning the gallery installations created through the project is rooted in ideas of 'the uncanny' and 'the sublime', postulated by philosophers such as Edmund Burke and Immanuel Kant, and demonstrated by artists such as Walter de Maria, Bruce Nauman, Cornelia Parker and James Turrell. From a scientific perspective, the project is clearly aimed at furthering ways of achieving public engagement and refining research already begun using STEM expertise and the ARISE project, based largely at the Meteorological Department of the University of Reading.

Although the article contains some minor grammatical typos throughout (no full-stop, end of 72; were/was, 113; capital To, 150; comma after 1st word, 182 etc.…) it is an interesting account of an innovative project and gives good information regarding its public presentation and outreach feedback. However, although the ongoing collaborative process is, in general, well documented, there is an implication within the article that the scientists are often problem-solving the artist`s practical needs, with no in-depth interrogation and analysis by the team of the visual, audible and physical aesthetic of the objects and installations generated by the overall process and how this informs the scientists` own research and insights. It would be interesting, for instance, to have (280-344) much more detail about how the collaborative process altered each member of the team`s initial ideas and approaches to his/her own subject. I believe that the article would also benefit from a more in-depth description of how and why these particular individuals from these specific disciplines began working collaboratively in the first place – what their original expectations were with regard to research -  and how they intend to incorporate aspects of the public feedback they gathered to develop this
**Fig. 1.**

Journal: GC
Title: Developing the hertz art-science project to allow inaudible sounds of the Earth and Cosmos to be experienced
Author(s): Graeme J. Marlton and Juliet Robson
MS No.: gc-2020-9
MS Type: Research article
Special Issue: Five years of Earth sciences and art at the EGU (2015–2019)

In essence, this article aims to record the creative collaborative process between and artist and a scientist by documenting the research development of a project and its dissemination, with a particular focus on public engagement and a lay-person`s interpretation of a potentially awe-inspiring science-based art installation.

The central pivot to the project is the use of **infrasound** to encourage individuals from diverse age groups and backgrounds to consider the continual 'invisible' movement and vibrations generated by natural and man-made activity within our planet – to reveal the imperceptible. This ambitious idea for an interactive installation is detailed by the authors as follows: *Drawing on the premise that everything vibrates, from the smallest atom to the furthest star, their frequencies surround us and yet leave no imprint, hertz would enable people to feel their bodies resonating to the inaudible symphony of our own planet and experience the stars singing and see their sound made visible. hertz's ultimate goal would aim to reconnect us to our planet and place in the cosmos.* (68-71)

From an artistic perspective, the conceptual structure underpinning the gallery installations created through the project is rooted in ideas of 'the uncanny' and 'the sublime', postulated by philosophers such as Edmund Burke and Immanuel Kant, and demonstrated by artists such as Walter de Maria, Bruce Nauman, Cornelia Parker and James Turrell. From a scientific perspective, the project is clearly aimed at furthering ways of achieving public engagement and refining research already begun using STEM expertise and the ARISE project, based largely at the Meteorological Department of the University of Reading.

Although the article contains some minor grammatical typos throughout (no full-stop, end of 72; were/was, 113; capital To, 150; comma after 1st word, 182 etc…. therefore needs full proof reading throughout) it is an interesting account of an innovative project and gives good information regarding its public presentation and outreach feedback. However, although the ongoing collaborative process is, in general, well documented, there is an implication within the article that the scientists are often problem-solving the artist`s practical needs, with no in-depth interrogation and analysis by the team of the visual, audible and physical aesthetic of the objects and installations generated by the overall process and how this informs the scientists` own research and insights. It would be interesting, for instance, to have (280-344) much more detail about how the collaborative process altered each member of the team`s initial ideas and approaches to his/her own subject. I believe that the article would also benefit from a more in-depth description of how and why these particular individuals from these specific disciplines began working collaboratively in the first place – what their original expectations were with regard to research -  and how they intend to incorporate aspects of the public feedback they gathered to develop this

**Fig. 2.**

---

## Referee Comment (RC3) · Sydney Lancaster (Referee) · 9 Apr 2020

The manuscript outlines the development and presentation of an installation artwork, the goal of which was to enable infrasound from various sources to be experienced in tactile and audible ways but the general public. Marlton and Robson focus their discussion on the process of their collaboration, from the first trials of software and hardware for the installation work, through to the public reception and tour of the project and its reception by several audiences in a variety of venues. Moreover, the presentation of processed infrasound in real time in touring locations of the project provides a tangible and immediate connection for the audience. Inaudible and unseen aspects of our plan-

etary environment, in effect, become 'real' and provoke both emotional and intellectual responses from the public, and often, a desire to learn more. This is an admirable and positive outcome to the project, and entirely relevant to the goals of Geoscience Communication. Moreover, the attention paid by the authors to the process of their constructive art-science collaboration is particularly relevant to the mission of Geoscience Communication, as it provides insights into the benefits and difficulties of such work for others interested in projects of this type.

This paper would benefit from a thorough proofreading for minor typos and some awkwardness in phrasing, but is generally readable and provides a solid overview of project development and the incorporation of feedback from public presentations.

Further, I feel the inclusion of more detailed information regarding the specifics of presenting the project is warranted. For example, it would be useful to know the volume (dB) at which the processed infrasound was presented; this is relevant both in terms of some of the negative responses (one of which was "scary," as conveyed by the authors), and in relation to the aspect of inclusivity/accessible design mentioned in the paper. It would be both instructive to those wishing to pursue a similar project, and informative to those seeking more detail with respect to accessibility - or simply practical considerations of venue - to include the details of all the hardware, software, and specifics (such as volume, mentioned above) in the paper, or in an appendix to it.

I would like to see more space devoted to the issues around accessible design overall, as this aspect of the project sets it apart from many art-science collaborations, and raises very important considerations in the transmission of both scientific and artistic/aesthetic information. Considerations around who our audiences are, and what are appropriate means of conveying ideas and information to them should be a first priority in this type of work, if we are to make inroads in communicating the relevance of both science and art to a wide audience. I commend the authors for raising this issue - but feel they could have addressed it more thoroughly, especially in relation to the user experience in the installation.

It would also be of benefit to contextualize the project further; framing hertz in relation to both research in infrasound and in the context of contemporary sound art would allow readers to better situate the project's relevance to developments in both disciplines, and highlight the benefits of such collaborations. Examples of this work may be found at Gupfinger, Ogawa, Sommerer, and Mignonneau (2009), Esquerro and Simon (2019), Sussman (2012), Hope (2009), Cranshaw (2014). In addition, there are other artists working with chladni plates; referencing their work would also assist in contextualizing the is aspect of the project, and strengthen the case for the relevance of this portion of the project here, and in future articles.

I commend the authors for including commentary on their own experiences of working collaboratively, across disciplines. This is challenging work, and can only be truly successful if everyone involved approaches the work with openness, and a desire to learn and work in new ways. There is tremendous value in this approach to the explication of both complex scientific concepts and artistic creation alike, and much to be learned on both 'sides.'

Specific comments related to the above are listed by line number here:

Line 93 The relationship (if any) between the data collected by the CTBTO and the data collected through the INFRA 20 is not clearly stated here.

Line 65 Although you are not detailing this part of the project here, it is part of your documentation of installations, and the statistics on visitor interaction with the works (Page 8, line 250). As such, would be useful to readers to cite and/or refer to work in cymatics, perhaps in particular reference to contemporary art, to further explicate the notion of making the invisible visible. The work of both Nigel Stanford and Gary James Joynes come to mind.

Line 120 -121 Could the sound emitted form the subwoofer also be felt physically? Worth noting one way or the other, as the secondary physical impact of the sound would contribute to the immersive quality. This seems to be the case given what you

say in the next sentence about playing Dark Side of the Moon through the subwoofer & transducer in the first trial, and with respect to the subwoofer in the public iterations of the project. More detailed specifications for the subwoofer and transducer, and dB for both initial test and subsequent installations would be extremely helpful.

Line 160 You could make more of the immediacy of the experience - it is an important factor in work of this nature that strives to connect people both emotionally and intellectually to natural phenomena such as this. This experience cannot be duplicated on the web, and cannot be simply listened to or watched: it needed people to be physically present. This becomes even more relevant in later iterations of the work, in which you draw on infrasound from the locations of presentation, where place and the experience of the work are inextricably linked.

Line 170 Was there feedback from the participants that was negative? Given the range of abilities in the audience for this, some may have had a negative experience; it would be useful to know this, what those less-positive responses were, and how they were factored in to further development of the project. For example, for some individuals with chronic pain and/or migraine and/or disabilities that affect balance, this installation may have been difficult to engage with, depending upon the volume or level of vibration physically experienced.

References Cited above:

Gupfinger, Reinhard & Ogawa, Hideaki & Sommerer, Christa & Mignonneau, Laurent. (2009). INTERACTIVE INFRASONIC ENVIRONMENT: A New Type of Sound Installation for Controlling Infrasound.

Ezquerro, L., and J. L. Simón. "Geomusic as a New Pedagogical and Outreach Resource: Interpreting Geoheritage with All the Senses." Geoheritage 11, no. 3 (September 1, 2019): 1187–98. https://doi.org/10.1007/s12371-019-00364-3.

Sussman,      M.      "Hearing      with      your      Body:      Infrasound.

https://www.artpractical.com/feature/hearing_with_your_body_infrasound/# Accessed 8 April 2020.

Hope, Cat. "Infrasonic Music." Leonardo Music Journal 2009 Vol. 19, 51-56.

Hope, Cat. "Earth pulse: Vibrational data as artistic inspiration." Re:Live Media Art histories 2009 Refereed Conference Proceedings (pp. 73-77), The University of Melbourne, 2009.

Crawshaw, Alexis Story. "Towards Defining the Potential of Electroacoustic Infrasonic Music." ICMC (2014).

Nigel Stanford. https://nigelstanford.com/Cymatics/Behind_the_Scenes.aspx

Gary James Jones. http://www.clinkersound.com/frequency-painting/?page_id=347

Please also note the supplement to this comment:
https://www.geosci-commun-discuss.net/gc-2020-9/gc-2020-9-RC3-supplement.pdf

---

## Referee Comment (RC4) · Anonymous Referee #4 · 14 Apr 2020

General comments

This paper highlights a fascinating project that brings together science and art in a strong collaboration. It should prove interesting to scientists with an interest in public engagement with research, as well as artists looking to draw on science. However, I think there are some weaknesses in the paper as it is presented, which make it difficult to follow and detract from what is otherwise interesting work. It is lacking in clarity at times and I believe it would benefit from more detail at certain points.

Specific comments

There are clear connections made to interesting and relevant science content through

the early stages of the project, during design and prototyping. The connection between research and the final installation is less clear, aside from the use of infrasound. Highlighting the scientific research content and how it was expressed in the main installation would substantially strengthen the connection to scientific questions, I think.

I would have appreciated an explication of the context of the project among related artworks. Have any other art installations used infrasound or is this the first? Are there other works that have used vibration in a similar or different way? How does this installation relate to other experiential works that incorporate scientific data? Similarly, explaining what other public engagement or artistic projects exist around infrasound or ARISE 2 would have helped site this work in the relevant landscape. Together, these would make much clearer the extent to which this work is novel.

Occasionally, the paper refers to 'playing the infrasound' (e.g. line 186, 187). I think this is a little disingenuous. The processing is quite carefully described, but my understanding of it from this is that what is actually played is synthetically generated pink noise, which is then processed according to the infrasound data. If my understanding is not quite right, then perhaps the section on the signal processing needs to be revisited. I wonder to what extent artistic licence was employed when creating the infrasound-scapes. Phrases such as 'This produced an effect that we felt was relatable to infrasound if we could hear it' suggest quite a lot, which in turn suggests a move away from the science. Perhaps the phrase 'keeping translatable authenticity' needs unpacking to clarify to what extent the experienced signals relate to the original infrasound signals. Relatedly, a flow diagram of sorts might help here (e.g. around line 152), to make clear exactly what the inputs, processes and outputs were for the prototypes and also for the final installation. Even after multiple readings of the paper, I'm still not sure I understand the relationship between the signals fed to the subwoofer and transducer – are they just the same?

The information on CTBTO stations and sensors is interesting, but I don't understand the connection between this and the project. Was data from these sensors used? Are

the microbarometers used by CTBTO the same as the microbarometer used on this project? The connection needs to be made clear; or if there is not one, this (lines 87-91) is probably extraneous and distracting information. Likewise, the reference to playing Pink Floyd through the system is confusing – did Pink Floyd use infrasound? Or was Pink Floyd's music used somewhere in the project? If there isn't any further connection, then I would suggest it is a distracting detail.

A point that is made in passing, but that I think deserves much more attention, is that 'you had to be physically present in order to sense the frequencies, making it an immersive and experiential artwork'. The fact that the artwork could not be reproduced through audio or video recording marks it out as something special in a world that seems increasingly focussed on engaging publics with research digitally. I think perhaps more could be made of this in the wider context of public engagement with research.

On the other hand, Section 4.4 on Web and online presence comes across as rather weak. Simply stating the numbers of impressions gives no context and no conclusion. Can any analysis be done of who the Twitter followers were or who visited the website? Were they scientists? Artists? Funders? What were the most popular posts and why? How does this performance compare to similar websites or accounts? The weakness of simply stating figures is noted in the text, but if nothing further can be added to this section by way of analysis, I would consider removing it. As it stands, I think it detracts from the flow of the paper.

The assessment of feedback from the tour was also somewhat underwhelming. It seems largely to consist of sharing positive comments. This section would be much stronger if this was better contextualised. How many comments were received? How many of those were positive / negative? Can the feedback be analysed in more detail? The word cloud seems like a good start, but are there themes to be drawn out? A clearer explanation of how this feedback impacted the project would also be beneficial. On a different note, I don't think the description of the installation as "scary" needs to

be considered negative, especially if part of the goal was to "re-establish links with the natural environment" including events that are "both majestic and alarming".

Much is made of the artist-scientist relationship in this work, and to my mind this (Section 5) is the weakest section of the paper; I would consider substantially reducing or rewriting it with a much tighter focus. A substantial portion of the text is devoted to expounding stereotypes about the differences between how scientists work and how artists work. This struck me as rather lazy writing. There are no citations of studies or research that look at this question, and I wonder what the basis is for these wide-ranging assertions about what scientists "will" do. Furthermore, as this paragraph progresses, it seems to lose its line of argument, and it is not clear what point is intended. Moreover, I would be wary of suggesting that the different ways two particular people react to a particular event (see lines 305-309) is as a result of one being an artist and the other a scientist – this is not a strong conclusion. Finally, the overall conclusion suggests that this collaboration "is a good model for future art science collaborations". To be more useful, I think the "model" in question needs further explanation. What was it they did that meant it worked especially well? What do other people need to know to be able to use the same model?

There are a number of grammatical and punctuation errors throughout the text that need fixing. I think it could also do with the attention of a copy editor to re-phrase a few passages as some of the writing is a little stilted. Amending these would substantially improve the readability.

Technical corrections: There were a couple of names and phrases that I think need explaining in the text. A few words of context would save me looking it up and give me a better frame of reference.

- Line 99, 'LT' is not explained – is this 'local time'?

- Line 165, what is the Attenborough Centre – is it an art space, a science space, a community space or something else?

- Line 211: What is 'We the Curious'?

There are some straightforward grammatical errors and misuses of punctuation

- Line 32: 'science technology engineering and maths' needs some commas

- Line 36: 'one of those was, co-author' – unnecessary comma

- Line 41: '(Wilson 1969) see figure 1' – needs some punctuation

- Line 58: 'the star in turn fluctuates in brightness, satellites like Kepler' – probably full stop, not comma

- Line 59: 'transiting exoplanet survey satellite' – this is the name of a particular satellite, treat it as such

- Line 98-99: inconsistency with spaces before 'Hz'

- Line 150: 'synthetic generated pink noise. to ensure' – capital T on 'to' Some sentences need re-phrasing, including

- Line 25: 'Technology further isolates the modern human from the natural environment in which we evolved increasingly being used as a filter through which we view the natural world.'

- Line 57: 'Sound waves move through sun stars gaseous interior because of temperature changes'

- Line 106: 'The infrasonic signals produced by the Reading thunderstorms and the infrasonic signal from the aurora is 4 times smaller.' Four times smaller than what?

- Line 107: 'This shows that different phenomena produce have different infrasound signatures'

- Line 113: 'Robson had a spare metal wheelchair made of metal that were good at transferring vibrations.' – intentional repetition of metal?

---

## Author Comment (AC1) · 22 Jun 2020

Reviewer 1: We thank the reviewer for their comments and have provided responses to those comments below.

Anonymous Referee

"The manuscript concerns a project which enabled infra sounds from various sources on Earth to be experienced through a multisensory artistic exhibit. The article focuses primarily on the development process of the exhibit, a collaboration between an artist

and scientists. This is relevant to the journal Geoscience Communication and would be of interest to those thinking of similar projects or approaching art-science collaborations. I do, however, have a number of concerns over the information presented which I detail here. Main issues: The introduction could do with much more of the broader context of science communication and public engagement that concerns this area of science or uses a similar method in order to properly frame this project. At present the motivations that people need to re-establish links with the natural environment come across as merely the opinions of the authors and not backed up by any published research or public dialogues. Only with this wider context is it possible to better consider the successes of this project."

In the revised version of this paper we will modify the introduction section to cite more references to back up the motivations and show how the hertz artwork will sit within the broader context For example: Gupfinger et al (2009), Esquerro and Simon (2019), Hope (2009), Cranshaw (2014).

"The main contribution that this article makes to the literature is arguably the development process of the exhibit. I applaud the authors for writing this in accessible way, however, interested technical readers may want more detailed information. I suggest the authors provide this in an appendix, e.g. giving precise parameters used in their processing so that others may be enabled to convert similar infrasound datasets."

We plan to include a technical appendix which details the filter coefficients used to create the infrasonic sound wave files.

"The evaluation data and its presentation in section 4 are rather lacking unfortunately. There is little to no detail of how "feedback" was collected, what specific questions were asked of participants, and how the qualitative data has been analysed. "

Due to the nature of the tour the co-authors were only present at the first event and were not able to oversee the data collection at the other venues in person. Thus, the feedback received was dependent on the venue in question, for example: At "We the

Curious" the quality of the feedback received was quite good. However, Tramway's feedback did not capture the public's feedback and only that of the organizers. Further to this feedback from the participants was entirely optional and the feedback cards left had little in the way of prompters. In hindsight it may have been better to devise 2 or 3 well defined questions to be asked on the exit of the exhibits.

"To this latter point the authors seem to have simply classified whether or not it was positive and provide, seemingly cherry-picked, example quotes. This work calls out for a thematic analysis to better understand what participants' responses to this experience were, what common themes emerged and how do they relate to the aims of the project and compare with other similar efforts? Can any conclusion be made linking back to the aims of the project, e.g. did it reconnect participants back with the Earth? "

Given the above highlighted issues with data quality we will rewrite the feedback section and perform a different analysis which would seek to answer, using the data available, Did participants feel more connected with the earth after interacting with the exhibits. This would be undertaken using a thematic approach as suggested by the reviewer.

"While the review of the collaboration is also interesting, more discussion and conclusions need to be drawn from the quotes provided."

The aim of this paper is to describe the motivations, implementation and feedback from the project. We included some detail on the collaboration with a small discussion. It is possible to go into greater depth about this and this may overshadow the work itself. We present our key findings from working together in the collaboration that others can use in a more in depth discussion in the area.

Specific comments: "Throughout the term "resonance" seems to be used slightly carelessly. It is not clear to this reviewer whether it is truly resonances which lead to many of the infrasounds considered (indeed many of them seem to be rather broadband rather than peaking at well defined frequencies), nor is it clear whether the transducers' vibrations are waves which are resonating within the human body. I would suggest the

authors consider carefully each usage of this word and only include it where appropriate (e.g. Its usage in describing asteroseismology is correct) and provide references, otherwise other terms such as sound, vibrations etc. should be used."

Yes- the reviewer is correct, resonance is when an object is made to oscillate or vibrate at its natural frequency We will check thoroughly through the text and use the correct scientific descriptions. I.e. vibration, oscillations, frequency etc.

"Line 98: Arguably the enhanced infrasound power goes to a much lower frequency than 0.1Hz in Figure 2, approximately 0.02Hz."

We will amend this in the revised manuscript

"Line 99: LT as Local Time needs to be introduced in the text."

We will amend this in the revised manuscript

"Line 105: There is no visible power enhancement at 1Hz in Figure 4, instead the biggest peaks appears to be around 0.03Hz."

This is a typo, we meant to say the power enhancement is below 1 Hz.

"Line 106: This sentence is confusing. You need to specify what quantity you are referring to exactly and whether you are comparing the two events to one another of the reference in the previous sentence."

We will reword this to highlight that the amplitude of the waveform is less than in figure 3 and the frequency bandwidth is also lower than in figure 3

"Line 123: "documented by smart phone" comes across as though the authors made notes using a smart phone, whereas I understand from later sentences they used an audio recording app on the phone. This should be made clearer."

We will elaborate on this in the revised manuscript: When performing tests we used a smart phone to video the participants response, nominally one of the co-authors,

interacting with the exhibit rather than attempting to record the exhibit itself.

"Line 137: It is not clear how the amplitude was measured and used to modulate tones. The authors may want to keep such technical detail to the suggested appendix though."

Here a sine wave with a given frequency (60Hz) was generated and then it was multiplied element wise by the band passed filtered infrasonic signal I(t) so that the played sound wave S: $S(t)=I(t)sin(2*pi*60*t)$

"Lines 143-160: Polyphonic seems to be the wrong term here, since this is defined as "a type of musical texture consisting of two or more simultaneous lines of independent melody" whereas the authors describe modulated pink noise which is not musical or melodious. What the authors describe is surely more of a cacophony than symphony. It would be very helpful to provide sound clips of the different processed versions of the infrasound for the readers to be able to interpret. Furthermore, on this point, the authors' descriptions of the sounds come across as a little hyperbolic and would benefit from some other viewpoints."

In the amended manuscript we will work on the wording describing the sound waves played. We will add a data repository that will include a sample audio clip that was played on the Radio 3 show so the curious reader can listen. (It should be noted the low pass filter is increased to allow it to be audible through conventional PC speakers)

"Line 167: "practicalities of access needs" are raised but no description or discussion of what these were are given."

More details about how disability access was addressed and incorporated into the project and the motivations behind it will be addressed in the revised Manuscript.

"Line 179-180: "further positive media coverage" is mentioned but no quotes or analysis of the material are presented."

We can include some quotes from the two references in the revised manuscript

"Lines 247-253: It needs to be stated how all of these were measured."

This data was collected by staff overseeing the exhibit at the venue. This will be amended in the revised manuscript.

"Lines 271-26: The numbers quoted here are rather meaningless without benchmarking against similar efforts. Furthermore, qualitative analysis of any tweets about the exhibit(not merely retweets or likes) could provide insight into audiences' responses, which is currently lacking."

This was an attempt to gauge the social media response online. It seems that this small section adds little and will be retracted in the revised manuscript

"Line 308: This should say Figure 1."

This will be amended in the revised manuscript

"Line 316: It is not clear who did the interviewing."

Marlton was interviewed by Robson as part of the required evaluation for Unlimited's grant evaluation. Robson was interviewed by Liz Hingly http://lizhingley.com/about , the project curator for https://www.phyartuob.co.uk. This will be added in the revised manuscript

References: Gupfinger, R., 2009, Interactive Infrasonic Environment: A new type of sound installation for controlling infrasound. In Workshop-Proceedings der Tagung Mensch & Computer 2009. Logos Verlag.

Ezquerro, L., and J. L. Simón.,2019, "Geomusic as a New Pedagogical and Outreach Resource: Interpreting Geoheritage with All the Senses." Geoheritage 11, no. 3, 1187–98. https://doi.org/10.1007/s12371-019-00364-3.Sussman,M.

Hope, C., 2009, "Infrasonic Music." Leonardo Music Journal 2009 Vol. 19, 51-56.

Crawshaw, A., 2014, "Towards Defining the Potential of Electroacoustic Infrasonic Music." ICMC

---

## Author Comment (AC2) · 22 Jun 2020

Reviewer 2: We thank the reviewer for their comments and respond to their comments below:

Charlie Hooker (Referee)charlie.hooker234@btinternet.com

Journal: GC Title: Developing the hertz art-science project to allow inaudible sounds of the Earth and Cosmos to be experienced Author(s): Graeme J. Marlton and Juliet Robson

[Figure]

MS No.: gc-2020-9

MS Type: Research article Special Issue: Five years ofEarth sciences and art at the EGU (2015–2019)

"In essence, this article aims to record the creative collaborative process between an artist and a scientist by documenting the research development of a project and its dissemination, with a particular focus on public engagement and a lay-person's interpretation of a potentially awe-inspiring science-based art installation. The central pivot to the project is the use of infrasound to encourage individuals from diverse age groups and backgrounds to consider the continual 'invisible' movement and vibrations generated by natural and man-made activity within our planet – to reveal the imperceptible. This ambitious idea for an interactive installation is detailed by the authors as follows: Drawing on the premise that everything vibrates, from the smallest atom to the furthest star, their frequencies surround us and yet leave no imprint, hertz would enable people to feel their bodies resonating to the inaudible symphony of our own planet and experience the stars singing and see their sound made visible. hertz's ultimate goal would aim to reconnect us to our planet and place in the cosmos. (68-71) From an artistic perspective, the conceptual structure underpinning the gallery installations created through the project is rooted in ideas of 'the uncanny' and 'the sublime', postulated by philosophers such as Edmund Burke and Immanuel Kant, and demonstrated by artists such as Walter de Maria, Bruce Nauman, Cornelia Parker and James Turrell. From a scientific perspective, the project is clearly aimed at furthering ways of achieving public engagement and refining research already begun using STEM expertise and the ARISE project, based largely at the Meteorological Department of the University of Reading. Although the article contains some minor grammatical typos throughout (no full-stop,end of 72; were/was, 113; capital To, 150; comma after 1st word, 182 etc....therefore needs full proof reading throughout) it is an interesting account of an innovative project and gives good information regarding its public presentation and outreach feedback."
[Figure]

A full proof reading of the revised manuscript will be undertaken before resubmission.

"However, although the ongoing collaborative process is, in general, well documented, there is an implication within the article that the scientists are often problem-solving the artist's practical needs, with no in-depth interrogation and analysis by the team of the visual, audible and physical aesthetic of the objects and installations generated by the overall process and how this informs the scientists' own research and insights. It would be interesting, for instance, to have (280-344) much more detail about how the collaborative process altered each member of the team's initial ideas and approaches to his/her own subject. I believe that the article would also benefit from a more in-depth description of how and why these particular individuals from these specific disciplines began working collaboratively in the first place – what their original expectations were with regard to research - and how they intend to incorporate aspects of the public feedback they gathered to develop this extremely interesting project further. This could be more fully developed in Section 6, where common methodologies (347) would benefit from being described in much more depth, to reveal the successes, failures and critical analysis of each discipline's methodologies and how the combined methodologies systematically achieved the final outcomes and, potentially, a new methodology. The article explores an intriguing topic, but could give a more rigorous record of the positive and negative surprises generated when two disciplines come together."

We will rewrite this section with emphasis looking at other art-science collaborations such as those described in Leach (2005) and in Webster (2005) compare to the project described here. We will also compare how the approach of Tsoupikova et al (2013) differed to ours. We will also draw similarities from Eldred (2016) who demonstrated how art collaboration can benefit problem solving especially for scientists. Whilst keeping this section concise and focused we will try and add more from the authors experiences to provide a fruitful account of the process for others to draw on.

"The art/science project itself seems to offer the public a potentially poetic experience and to be physically engaging. However, from the article, I do not quite understand

**[GCD](GCD)**
the gallery context of the immersive audience participation. If I walked into the gallery, exactly what would lead me to sit on the chair and how would I understand the implications of the uncanny and mysterious source that I was listening to? Is it a feature of the work that there should always be information sheets or 'exhibit demonstrators' available for the public, or does the installation reveal its meaning in another more subtle way – more akin to, say, a Joseph Beuys installation? The article would therefore benefit from a passage describing the team's views regarding public engagement methodology – the pros and cons of installing an object which emanates a scientific principle through its construction and physical location without the principle needing to be contextualised by an additional means - as this does not appear to be fully documented or analysed.

Professor Charlie Hooker."

Hertz was firstly exhibited at the Oxford IF festival with an educational agenda as this was the context of the science festival and where the emphasis lay. Marlton and Robson were all present and each gave talks about their involvement, how the piece worked and their individual research after which people were invited to interact with the infrasound machines. The audience also had the opportunity to talk to all the collaborators present.

For the exhibitions at Tramway and We The Curious a more sensorial, experiential encounter that emphasised hertz's aesthetic and artistic aspects was pursued. Figure 7 of the paper shows an image of the hertz piece and how it was set up at the Tramway in Glasgow. In section 2 of the paper we described how the subwoofer was used to fill a space with the loud low frequency sounds that were also passed through the vibrating furniture. It was envisaged that sound from the subwoofer would entice people towards the exhibit. Due to a limited budget, talks by the co-authors at We The Curious and the Tramway could not be undertaken. Therefore at Tramway in consultation with the curator an interpretation board was used with a short introduction to the work and invigilators would be briefed and available to answer questions if asked. Unfortunately, since there was a lack of feedback from the public at the venue it is difficult to know

how well this worked.

The approach for We The Curious was for the roving educational team to be briefed on hertz and for the project to be included in educational demonstrations of exhibits at We The Curious when they happened. We The Curious is a science venue with a dedicated space (The Box) for artworks and hertz was the venues first commissioned piece. More scientific information was included in the interpretation and postcards with relevant images and brief facts about infrasound were available. Invigilators were also briefed with information on the project to enable them to answer questions.

In a revised manuscript we will describe in more detail in section 3 how the piece was curated at each venue and include much of the above discussion there.

References Eldred, S. M. (2016). Art–science collaborations: Change of perspective. Nature, 537(7618), 125-126. – How working on a art project changes your perceptions of your own work

Leach, J. (2005). 'Being in Between': Art-Science Collaborations and a Technological Culture. Social Analysis, 49(1), 141-162.

Webster, S. (2005). Art and science collaborations in the United Kingdom. Nature Reviews Immunology, 5(12), 965-969.

Tsoupikova, D., Kostis, H. N., & Sandin, D. (2013). A practical guide to art/science collaborations. In ACM SIGGRAPH 2013 Courses (pp. 1-55).

---

## Author Comment (AC3) · 22 Jun 2020

Reviewer 3: We thank the Referee for their comments and have responded below.

Sydney Lancaster (Referee) sydney.lancaster@gmail.com

"The manuscript outlines the development and presentation of an installation artwork, the goal of which was to enable infrasound from various sources to be experienced in tactile and audible ways but the general public. Marlton and Robson focus their discussion on the process of their collaboration, from the first trials of software and hardware

for the installation work, through to the public reception and tour of the project and its reception by several audiences in a variety of venues. Moreover, the presentation of processed infrasound in real time in touring locations of the project provides a tangible and immediate connection for the audience. Inaudible and unseen aspects of our planetary environment, in effect, become 'real' and provoke both emotional and intellectual responses from the public, and often, a desire to learn more. This is an admirable and positive outcome to the project, and entirely relevant to the goals of Geoscience Communication. Moreover, the attention paid by the authors to the process of their constructive art-science collaboration is particularly relevant to the mission of Geoscience Communication, as it provides insights into the benefits and difficulties of such work for others interested in projects of this type. This paper would benefit from a thorough proofreading for minor typos and some awkwardness in phrasing but is generally readable and provides a solid overview of project development and the incorporation of feedback from public presentations."

We will thoroughly proofread through the revised manuscript.

"Further, I feel the inclusion of more detailed information regarding the specifics of presenting the project is warranted. For example, it would be useful to know the volume(dB) at which the processed infrasound was presented; this is relevant both in terms of some of the negative responses (one of which was "scary," as conveyed by the authors), and in relation to the aspect of inclusivity/accessible design mentioned in the paper. It would be both instructive to those wishing to pursue a similar project, and informative to those seeking more detail with respect to accessibility - or simply practical considerations of venue - to include the details of all the hardware, software, and specifics (such as volume, mentioned above) in the paper, or in an appendix to it."

A technical appendix will be added to the revised manuscript detailing the amplitude of the processed infrasound and some of the filter coefficients used. More information can be added about the hardware used. I would like to see more space devoted to the issues around accessible design overall, as this aspect of the project sets it

apart from many art-science collaborations, and raises very important considerations in the transmission of both scientific and artistic/aesthetic information. Considerations around who our audiences are, and what are appropriate means of conveying ideas and information to them should be a first priority in this type of work, if we are to make inroads in communicating the relevance of both science and art to a wide audience. I commend the authors for raising this issue – but feel they could have addressed it more thoroughly, especially in relation to the user experience in the installation."

We will be including at the end of section 3 more information about how the science is conveyed at each venue and the setup of the exhibit. In this section we will also discuss the disability access incorporated into the project and the motivations behind it.

"It would also be of benefit to contextualize the project further; framing hertz in relation to both research in infrasound and in the context of contemporary sound art would allow readers to better situate the project's relevance to developments in both disciplines, and highlight the benefits of such collaborations. Examples of this work may be found at: Gupfinger, Ogawa, Sommerer, and Mignonneau (2009), Esquerro and Simon (2019),Sussman (2012), Hope (2009), Cranshaw (2014). In addition, there are other artists working with chladni plates; referencing their work would also assist in contextualizing the is aspect of the project, and strengthen the case for the relevance of this portion of the project here, and in future articles."

The introduction section will be modified to contextualise the project further with references drawn from artists working with both chladni patterns and infrasound previous to the project and those discovered during the research phase while the project was ongoing. In addition to this references to other pieces which have made similar use a geophysical data and those mentioned by the reviewer above will be used in the introduction section to better seat the project in a broader context in the art-science world.

"I commend the authors for including commentary on their own experiences of working collaboratively, across disciplines. This is challenging work, and can only be truly successful if everyone involved approaches the work with openness, and a desire to learn and work in new ways. There is tremendous value in this approach to the explication of both complex scientific concepts and artistic creation alike, and much to be learned on both 'sides.'

Specific comments related to the above are listed by line number here:

Line 93 The relationship (if any) between the data collected by the CTBTO and the data collected through the INFRA 20 is not clearly stated here."

The mention of the CTBTO was to give a broader context to why infrasound data was recorded originally and data from the CTBTO was not explicitly used in the artwork. The revised manuscript will be amended to clarify this.

"Line 65 Although you are not detailing this part of the project here, it is part of your documentation of installations, and the statistics on visitor interaction with the works(Page 8, line 250). As such, would be useful to readers to cite and/or refer to work in cymatics, perhaps in particular reference to contemporary art, to further explicate the notion of making the invisible visible. The work of both Nigel Stanford and Gary James Joynes come to mind."

As discussed above we plan to contextualise and sit the project in a broader context with references to other artists in the introduction in the revised manuscript

"Line 120 -121 Could the sound emitted form the subwoofer also be felt physically? Worth noting one way or the other, as the secondary physical impact of the sound would contribute to the immersive quality. This seems to be the case given what you say in the next sentence about playing Dark Side of the Moon through the subwoofer & transducer in the first trial, and with respect to the subwoofer in the public iterations of the project. More detailed specifications for the subwoofer and transducer, and dB

for both initial test and subsequent installations would be extremely helpful."

Yes the sound from the subwoofer could be felt physically. The dB was never explicitly measured nor calculated. The information about the power of the subwoofer and transducer will give some indicator to the loudness. It should be further noted that volume levels were shifted from install site to install site.

"Line 160 You could make more of the immediacy of the experience - it is an important factor in work of this nature that strives to connect people both emotionally and intellectually to natural phenomena such as this. This experience cannot be duplicated on the web, and cannot be simply listened to or watched: it needed people to be physically present. This becomes even more relevant in later iterations of the work, in which you draw on infrasound from the locations of presentation, where place and the experience of the work are inextricably linked."

We will make this aspect more explicit in the revised manuscript. By reaffirming that in order to experience it truly one had to visit the exhibit and that even processed infrasound data that had a higher threshold so it could be listened to through standard audio equipment did not do the project justice

"Line 170 Was there feedback from the participants that was negative? Given the range of abilities in the audience for this, some may have had a negative experience; it would be useful to know this, what those less-positive responses were, and how they were factored in to further development of the project. For example, for some individuals with chronic pain and/or migraine and/or disabilities that affect balance, this installation may have been difficult to engage with, depending upon the volume or level of vibration physically experienced."

Section 3 will be modified to describe more fully the integration of accessibility into the design of hertz as well as feedback from those who experience accessibility challenges. Visually impaired and deaf visitors (figure 6 of the manuscript) at the "Be There At the Start" conference found it easy to interact with the project. While there were a

wide range of people with different disabilities at the conference, many of which experienced the infrasound of Leicester through the prototype of hertz there was no negative feedback in terms of discomfort. Robson experiences chronic pain and had found no ill effects, some vibrations were soothing and some intense but did not exacerbate her chronic pain.

"References Cited above: Gupfinger, Reinhard & Ogawa, Hideaki & Sommerer, Christa & Mignonneau, Laurent.(2009). INTERACTIVE INFRASONIC ENVIRONMENT: A New Type of Sound Installation for Controlling Infrasound.

Ezquerro, L., and J. L. Simón. "Geomusic as a New Pedagogical and Outreach Re-source: Interpreting Geoheritage with All the Senses." Geoheritage 11, no. 3 (Septem-ber 1, 2019): 1187–98. https://doi.org/10.1007/s12371-019-00364-3.Sussman,M. "HearingwithyourBody:Infrasound https://www.artpractical.com/feature/hearing_with_your_body_infrasound/# Accessed8 April 2020.Hope, Cat. "Infrasonic Music." Leonardo Music Journal 2009 Vol. 19, 51-56.

Hope, Cat. "Earth pulse: Vibrational data as artistic inspiration." Re:Live Media Arthistories 2009 Refereed Conference Proceedings (pp. 73-77), The University of Melbourne, 2009.

Crawshaw, Alexis Story. "Towards Defining the Potential of Electroacoustic Infrasonic-Music." ICMC (2014).

Nigel Stanford. https://nigelstanford.com/Cymatics/Behind_the_Scenes.aspxGary James Jones. http://www.clinkersound.com/frequency-painting/?page_id=347

Please also note the supplement to this comment:https://www.geosci-commun-discuss.net/gc-2020-9/gc-2020-9-RC3-supplement.pdfInteractive comment on Geosci. Commun. Discuss., https://doi.org/10.5194/gc-2020-9, 2020.C5

---

## Author Comment (AC4) · 22 Jun 2020

Referee 4: We thank the reviewer for their comments and respond to their comments below:

Anonymous Referee #4

"General comments: This paper highlights a fascinating project that brings together science and art in a strong collaboration. It should prove interesting to scientists with an interest in public engagement with research, as well as artists looking to draw on

science. However, I think there are some weaknesses in the paper as it is presented, which make it difficult to follow and detract from what is otherwise interesting work. It is lacking in clarity at times and I believe it would benefit from more detail at certain points. Specific comments: There are clear connections made to interesting and relevant science content through the early stages of the project, during design and prototyping. The connection between research and the final installation is less clear, aside from the use of infrasound. High-lighting the scientific research content and how it was expressed in the main installation would substantially strengthen the connection to scientific questions, I think. I would have appreciated an explication of the context of the project among related artworks. Have any other art installations used infrasound or is this the first? Are there other works that have used vibration in a similar or different way? How does this installation relate to other experiential works that incorporate scientific data? Similarly, explaining what other public engagement or artistic projects exist around infrasound or ARISE 2 would have helped site this work in the relevant landscape."

The introduction section is to be amended to include references to similar artworks and their artists, with aims to contextualise the artwork in a broader context.

"Together, these would make much clearer the extent to which this work is novel. Occasionally, the paper refers to 'playing the infrasound' (e.g. line 186, 187). I think this is a little disingenuous. The processing is quite carefully described, but my under-standing of it from this is that what is actually played is synthetically generated pinknoise, which is then processed according to the infrasound data. If my understanding is not quite right, then perhaps the section on the signal processing needs to be revisited. I wonder to what extent artistic licence was employed when creating the infrasound-scapes. Phrases such as 'This produced an effect that we felt was relatable to infrasound if we could hear it' suggest quite a lot, which in turn suggests a move away from the science. Perhaps the phrase 'keeping translatable authenticity' needs unpacking to clarify to what extent the experienced signals relate to the original infrasound signals.

Interactive
comment

[Figure]

Relatedly, a flow diagram of sorts might help here (e.g. aroundline 152), to make clear exactly what the inputs, processes and outputs were for the prototypes and also for the final installation. Even after multiple readings of the paper, I'm still not sure I understand the relationship between the signals fed to the subwoofer and transducer – are they just the same?"

The author is correct. The artificially generated pink noise amplitude is modulated by the amplitude of the band pass filtered infrasound. Scientific license is added by setting the band pass filter bands based on the frequency domain of the detected infrasound which is shown in figures 3 & 4. This ensures that only the band pass filtered signal is used to modulate the pink noise. The modulated pink noise was then low passed filtered to only enable the low frequency parts of pink noise to be played via the PCs sound card to the subwoofer and transducer. To clarify this we will adapt figure 5 to act as both a diagram and flowchart.

"The information on CTBTO stations and sensors is interesting, but I don't understand the connection between this and the project. Was data from these sensors used? Are the microbarometers used by CTBTO the same as the microbarometer used on this project? The connection needs to be made clear; or if there is not one, this (lines87-91) is probably extraneous and distracting information."

The inclusion of the CTBTO was to include some context as to why and how infrasound is monitored across the globe to give the reader some additional background. The data detected by our sensor is effectively the same kind of data collected by the CTBTO sensors, but isn't included in the project itself. This section will be reworded to clarify this.

"Likewise, the reference to playing Pink Floyd through the system is confusing – did Pink Floyd use infrasound? Or was Pink Floyd's music used somewhere in the project? If there isn't any further connection, then I would suggest it is a distracting detail. "

It was a sound file used as a to initially test the system. We will retract these lines

"A point that is made in passing, but that I think deserves much more attention, is that 'you had to be physically present in order to sense the frequencies, making it an immersive and experiential artwork'. The fact that the artwork could not be reproduced through audio or video recording marks it out as something special in a world that seems increasingly focussed on engaging publics with research digitally. I think perhaps more could be made of this in the wider context of public engagement with research. On the other hand, Section 4.4 on Web and online presence comes across as rather weak. Simply stating the numbers of impressions gives no context and no conclusion. Can any analysis be done of who the Twitter followers were or who visited the website? Were they scientists? Artists? Funders? What were the most popular posts and why? How does this performance compare to similar websites or accounts? The weakness of simply stating figures is noted in the text, but if nothing further can be added to this section by way of analysis, I would consider removing it. As it stands, I think it detracts from the flow of the paper."

We will remove this and section 4 (the feedback section) will be refocused to look at all available feedback from across all venues to assess whether hertz met is aims to reconnect people with the Earth.

"The assessment of feedback from the tour was also somewhat underwhelming. It seems largely to consist of sharing positive comments. This section would be much stronger if this was better contextualised. How many comments were received? How many of those were positive / negative? Can the feedback be analysed in more detail? The word cloud seems like a good start, but are there themes to be drawn out? A clearer explanation of how this feedback impacted the project would also be beneficial. On a different note, I don't think the description of the installation as "scary" needs to be considered negative, especially if part of the goal was to "re-establish links with the natural environment" including events that are "both majestic and alarming"."

Due to the nature of the tour the co-authors were only present at the first event and were not able to oversee the data collection at the other venues in person. Thus the

feedback received was dependent on the venue in question, for example: At "We the Curious" the quality of the feedback received was quite good. However, Tramway's feedback did not capture the public's feedback and only that of the organisers. Further to this feedback from the participants was entirely optional and the feedback cards left had little in the way of prompters. In hindsight it may have been better to devise 2 or 3 well defined questions to be asked on the exit of the exhibits. Given the above highlighted issues with data quality we will rewrite the feedback section and perform a different analysis which would seek to answer, using the data available, Did participants feel more connected with the earth after interacting with the exhibits. This would be undertaken using a thematic approach as suggested by the reviewer

"Much is made of the artist-scientist relationship in this work, and to my mind this (Section 5) is the weakest section of the paper; I would consider substantially reducing or rewriting it with a much tighter focus. A substantial portion of the text is devoted to expounding stereotypes about the differences between how scientists work and how artists work. This struck me as rather lazy writing. There are no citations of studies or research that look at this question, and I wonder what the basis is for these wide-ranging assertions about what scientists "will" do. Furthermore, as this paragraph progresses, it seems to lose its line of argument, and it is not clear what point is intended. Moreover, I would be wary of suggesting that the different ways two particular people react to a particular event (see lines 305-309) is as a result of one being an artist and the other a scientist – this is not a strong conclusion. Finally, the overall conclusion suggests that this collaboration "is a good model for future art science collaborations". To be more useful, I think the "model" in question needs further explanation. What was it they did that meant it worked especially well? What do other people need to know to be able to use the same model?"

We will rewrite this section with emphasis looking at other art-science collaborations such as those described in Leach (2005) and in Webster (2005) compare to the project described here. We will also compare how the approach used here differed from that of

Tsoupikova et al (2013) We will also draw similarities from Eldred (2016) who demonstrated how art collaboration can benefit problem solving especially for scientists.

"There are a number of grammatical and punctuation errors throughout the text that need fixing. I think it could also do with the attention of a copy editor to re-phrase a few passages as some of the writing is a little stilted. Amending these would substantially improve the readability. Technical corrections: There were a couple of names and phrases that I think need explaining in the text. A few words of context would save me looking it up and give me a better frame of reference.- Line 99, 'LT' is not explained – is this 'local time'?-"

Yes we will add this

"Line 165, what is the Attenborough Centre – is it an art space, a science space, acommunity space or something else?"

It is, we will clarify this in the revised manuscript

"Line 211: What is 'We the Curious'?" It is an educational science gallery We will clarify this in the revised manuscript.

"There are some straightforward grammatical errors and misuses of punctuation- Line 32: 'science technology engineering and maths' needs some commas- Line 36: 'one of those was, co-author' – unnecessary comma- Line 41: '(Wilson 1969) see figure 1' – needs some punctuation- Line 58: 'the star in turn fluctuates in brightness, satellites like Kepler' – probably fullstop, not comma- Line 59: 'transiting exoplanet survey satellite' – this is the name of a particular satellite,treat it as such- Line 98-99: inconsistency with spaces before 'Hz'- Line 150: 'synthetic generated pink noise. to ensure' – capital T on 'to' Some sentences need re-phrasing, including- Line 25: 'Technology further isolates the modern human from the natural environmentin which we evolved increasingly being used as a filter through which we view thenatural world.'- Line 57: 'Sound waves move through sun stars gaseous interior because of temper-ature changes'-

Line 106: 'The infrasonic signals produced by the Reading thunderstorms and the infrasonic signal from the aurora is 4 times smaller.' Four times smaller than what?- Line 107: 'This shows that different phenomena produce have different infrasound signatures' Line 113: 'Robson had a spare metal wheelchair made of metal that were good at transferring vibrations.' – intentional repetition of metal?"

These will be addressed and changed in the revised manuscript

References Eldred, S. M. (2016). Art–science collaborations: Change of perspective. Nature, 537(7618), 125-126. – How working on a art project changes your perceptions of your own work

Leach, J. (2005). 'Being in Between': Art-Science Collaborations and a Technological Culture. Social Analysis, 49(1), 141-162.

Webster, S. (2005). Art and science collaborations in the United Kingdom. Nature Reviews Immunology, 5(12), 965-969.

Tsoupikova, D., Kostis, H. N., & Sandin, D. (2013). A practical guide to art/science collaborations. In ACM SIGGRAPH 2013 Courses (pp. 1-55).

---

## Author Response (AR1)

We thank the reviewers for their comments and have provided responses to those comments below. Changes to the manuscript have been highlighted in red at the end of the rebuttal.

*Reviewer 1:*

*Anonymous Referee #1

*The manuscript concerns a project which enabled infra sounds from various sources on Earth to be experienced through a multisensory artistic exhibit. The article focuses primarily on the development process of the exhibit, a collaboration between an artist and scientists. This is relevant to the journal Geoscience Communication and would be of interest to those thinking of similar projects or approaching art-science collaborations. I do, however, have a number of concerns over the information presented which I detail here.*

*Main issues: The introduction could do with much more of the broader context of science communication and public engagement that concerns this area of science or uses a similar method in order to properly frame this project. At present the motivations that people need to re-establish links with the natural environment come across as merely the opinions of the authors and not backed up by any published research or public dialogues. Only with this wider context is it possible to better consider the successes of this project.*

In the revised version of this paper we have modified the introduction section to cite more references to show how the hertz artwork will sit within the broader context with other artworks that aim to make the intangible tangible. We have also included examples of other artworks including infrasound.

*The main contribution that this article makes to the literature is arguably the development process of the exhibit. I applaud the authors for writing this in accessible way, however, interested technical readers may want more detailed information. I suggest the authors provide this in an appendix, e.g. giving precise parameters used in their processing so that others may be enabled to convert similar infrasound datasets.*

A technical appendix has been added as supplementary material. This appendix includes specifics on the filters used and gives an idea over which frequency bands the filters were applied. Due to the nature of the project different cut-off frequencies in the band pass filtering were used dependent on source so the values quoted here are for a typical configuration.

*The evaluation data and its presentation in section 4 are rather lacking unfortunately. There is little to no detail of how "feedback" was collected, what specific questions were asked of participants, and how the qualitative data has been analysed.*

Due to the nature of the tour the co-authors were only present at the first event and were not able to oversee the data collection at the other venues in person. Thus, the feedback received was dependent on the venue in question, for example: At "We the Curious" the quality of the feedback received was quite good. However, Tramway's feedback did not capture the public's feedback and only that of the organizers.

Further to this feedback from the participants was entirely optional and the feedback cards left had little in the way of prompters. In hindsight it may have been better to devise 2 or 3 well defined questions to be asked on the exit of the exhibits.

*To this latter point the authors seem to have simply classified whether or not it was positive and provide, seemingly cherry-picked, example quotes. This work calls out for a thematic analysis to better understand what participants' responses to this experience were, what common themes emerged and how do they relate to the aims of the project and compare with other similar efforts? Can any conclusion be made linking back to the aims of the project, e.g. did it reconnect participants back with the Earth?*

Given the above highlighted issues with data quality we have re-written the feedback section. Participant feedback from We the Curious has been used to answer the question 'Did participants feel more connected with the earth after interacting with the exhibits?' We then perform a thematic analysis in Table 3 using the written feedback gathered during the project to explore the nature of this connection.

*While the review of the collaboration is also interesting, more discussion and conclusions need to be drawn from the quotes provided.*

This section has been re written, we have given the section more structure and have related our process through the project with current generalizations or other collaborations and have highlighted where these differ or are similar. We have also included how the project has improved or changed our working practices.

*Specific comments:*

*Throughout the term "resonance" seems to be used slightly carelessly. It is not clear to this reviewer whether it is truly resonances which lead to many of the infrasounds considered (indeed many of them seem to be rather broadband rather than peaking at well defined frequencies), nor is it clear whether the transducers' vibrations are waves which are resonating within the human body. I would suggest the authors consider carefully each usage of this word and only include it where appropriate (e.g. Its usage in describing asteroseismology is correct) and provide references, otherwise other terms such as sound, vibrations etc. should be used.*

Yes- the reviewer is correct; resonance is when an object is made to oscillate or vibrate at its natural frequency We have checked thoroughly through the text and have used the correct scientific descriptions. I.e. vibration, oscillations, frequency etc.

*Line 98: Arguably the enhanced infrasound power goes to a much lower frequency than 0.1Hz in Figure 2, approximately 0.02Hz.*

This has been changed

*Line 99: LT as Local Time needs to be introduced in the text.*

This has been introduced

*Line 105: There is no visible power enhancement at 1Hz in Figure 4, instead the biggest peaks appears to be around 0.03Hz.*

This sentence has been rephrased.

*Line 106: This sentence is confusing. You need to specify what quantity you are referring to exactly and whether you are comparing the two events to one another of the reference in the previous sentence.*

This sentence has been reworded to provide more clarity

*Line 123: "documented by smart phone" comes across as though the authors made notes using a smart phone, whereas I understand from later sentences they used an audio recording app on the phone. This should be made clearer.*

We have elaborated on this in the revised manuscript: When performing tests, we used a smart phone to video the participants response, nominally one of the co-authors, interacting with the exhibit rather than attempting to record the exhibit itself

*Line 137: It is not clear how the amplitude was measured and used to modulate tones. The authors may want to keep such technical detail to the suggested appendix though.*

A mathematical description of this has been added to manuscript

*Lines 143-160: Polyphonic seems to be the wrong term here, since this is defined as "a type of musical texture consisting of two or more simultaneous lines of independent melody" whereas the authors describe modulated pink noise which is not musical or melodious. What the authors describe is surely more of a cacophony than symphony. It would be very helpful to provide sound clips of the different processed versions of the infrasound for the readers to be able to interpret. Furthermore,*

*on this point, the authors' descriptions of the sounds come across as a little hyperbolic and would benefit from some other viewpoints.*

In the amended manuscript we have changed the wording describing the sound waves played from polyphonic to a cacophony. A a data repository that will include a sample audio clips that was played on the Radio 3 show so the curious reader can listen. (It should be noted the cut off frequency on the low pass filter is increased to allow it to be audible through conventional PC speakers) A gap for a DOI link has been left in the manuscript for this

*Line 167: "practicalities of access needs" are raised but no description or discussion of what these were are given.*

Section 4 discusses access consideration for the duration of the project

*Line 179-180: "further positive media coverage" is mentioned but no quotes or analysis of the material are presented.*

Quotes from the reference media sources have been included

*Lines 247-253: It needs to be stated how all of these were measured.*

This data was collected by staff overseeing the exhibit at the venue. This has been amended in the revised manuscript

*Lines 271-26: The numbers quoted here are rather meaningless without benchmarking against similar efforts. Furthermore, qualitative analysis of any tweets about the exhibit (not merely retweets or likes) could provide insight into audiences' responses, which is currently lacking.*

This section has been removed

*Line 308: This should say Figure 1.*

This has been amended in the revised manuscript

*Line 316: It is not clear who did the interviewing.*

Marlton was interviewed by Robson as part of the required evaluation for Unlimited's grant evaluation. Robson was interviewed by Liz Hingly http://lizhingley.com/about , the project curator for https://www.phyartuob.co.uk. This has been added in the revised manuscript

**Reviewer 2:**

We thank the reviewer for their comments and respond to their comments below:

*Charlie Hooker (Referee)charlie.hooker234@btinternet.com*

*Journal: GC Title: Developing the hertz art-science project to allow inaudible sounds of the Earth and Cosmos to be experienced Author(s): Graeme J. Marlton and Juliet Robson*

*MS No.: gc-2020-9*

*MS Type: Research article Special Issue: Five years of Earth sciences and art at the EGU (2015–2019)*

*In essence, this article aims to record the creative collaborative process between an artist and a scientist by documenting the research development of a project and its dissemination, with a particular focus on public engagement and a lay-person's interpretation of a potentially awe-inspiring science-based art installation. The central pivot to the project is the use of infrasound to encourage individuals from diverse age groups and backgrounds to consider the continual 'invisible' movement and vibrations generated by natural and man-made activity within our planet – to reveal the imperceptible. This ambitious idea for an interactive installation is detailed by the authors as follows: Drawing on the premise that everything vibrates, from the smallest atom to the furthest star, their frequencies surround us and yet leave no imprint, hertz would enable people to feel their bodies resonating to the inaudible symphony of our own planet and experience the stars singing and see their sound made visible. hertz's ultimate goal would aim to reconnect us to our planet and place in the cosmos.*

*(68-71) From an artistic perspective, the conceptual structure underpinning the gallery installations created through the project is rooted in ideas of 'the uncanny' and 'the sublime', postulated by philosophers such as Edmund Burke and Immanuel Kant, and demonstrated by artists such as Walter de Maria, Bruce Nauman, Cornelia Parker and James Turrell. From a scientific perspective, the project is clearly aimed at furthering ways of achieving public engagement and refining research already begun using STEM expertise and the ARISE project, based largely at the Meteorological Department of the University of Reading. Although the article contains some minor grammatical typos throughout (no full-stop, end of 72; were/was, 113; capital To, 150; comma after 1st word, 182 etc....therefore needs full proof reading throughout) it is an interesting account of an innovative project and gives good information regarding its public presentation and outreach feedback.*

A full proof reading of the revised manuscript has been undertaken.

*However, although the ongoing collaborative process is, in general, well documented, there is an implication within the article that the scientists are often problem-solving the artist's practical needs, with no in-depth interrogation and analysis by the*

*team of the visual, audible and physical aesthetic of the objects and installations generated by the overall process and how this informs the scientists' own research and insights. It would be interesting, for instance, to have (280-344) much more detail about how the collaborative process altered each member of the team's initial ideas and approaches to his/her own subject. I believe that the article would also benefit from a more in-depth description of how and why these particular individuals from these specific disciplines began working collaboratively in the first place – what their original expectations were with regard to research - and how they intend to incorporate aspects of the public feed-back they gathered to develop this extremely interesting project further. This could be more fully developed in Section 6, where common methodologies (347) would benefit from being described in much more depth, to reveal the successes, failures and critical analysis of each discipline's methodologies and how the combined methodologies systematically achieved the final outcomes and, potentially, a new methodology. The article explores an intriguing topic, but could give a more rigorous record of the positive and negative surprises generated when two disciplines come together.*

This section has been re written, we have given the section more structure and have related our process through the project with current generalizations or other collaborations and have highlighted where these differ or are similar. We have also included how the project has improved or changed our working practices.

*The art/science project itself seems to offer the public a potentially poetic experience and to be physically engaging. However, from the article, I do not quite understand the gallery context of the immersive audience participation. If I walked into the gallery, exactly what would lead me to sit on the chair and how would I understand the implications of the uncanny and mysterious source that I was listening to? Is it a feature of the work that there should always be information sheets or 'exhibit demonstrators' available for the public, or does the installation reveal its meaning in another more subtle way – more akin to, say, a Joseph Beuys installation? The article would therefore benefit from a passage describing the team's views regarding public engagement methodology – the pros and cons of installing an object which emanates a scientific principle through its construction and physical location without the principle needing to be contextualised by an additional means - as this does not appear to be fully documented or analysed.*

*Professor Charlie Hooker.*

In the revised manuscript we have described in more detail in section 5 how the piece was curated at each venue.

**Reviewer 3:**

*Sydney Lancaster (Referee) sydney.lancaster@gmail.com*

*The manuscript outlines the development and presentation of an installation artwork, the goal of which was to enable infrasound from various sources to be experienced in tactile and audible ways but the general public. Marlton and Robson focus their discussion on the process of their collaboration, from the first trials of software and hardware for the installation work, through to the public reception and tour of the project and its reception by several audiences in a variety of venues. Moreover, the presentation of processed infrasound in real time in touring locations of the project provides a tangible and immediate connection for the audience. Inaudible and unseen aspects of our planetary environment, in effect, become 'real' and provoke both emotional and intellectual responses from the public, and often, a desire to learn more. This is an admirable and positive outcome to the project, and entirely relevant to the goals of Geoscience Communication. Moreover, the attention paid by the authors to the process of their constructive art-science collaboration is particularly relevant to the mission of Geoscience Communication, as it provides insights into the benefits and difficulties of such work for others interested in projects of this type.*

*This paper would benefit from a thorough proofreading for minor typos and some awkwardness in phrasing but is generally readable and provides a solid overview of project development and the incorporation of feedback from public presentations.*

A thorough proofread through the revised manuscript has been undertaken

*Further, I feel the inclusion of more detailed information regarding the specifics of presenting the project is warranted. For example, it would be useful to know the volume(dB) at which the processed infrasound was presented; this is relevant both in terms of some of the negative responses (one of which was "scary," as conveyed by the authors), and in relation to the aspect of inclusivity/accessible design mentioned in the paper. It would be both instructive to those wishing to pursue a similar project, and informative to those seeking more detail with respect to accessibility - or simply practical considerations of venue - to include the details of all the hardware, software, and specifics (such as volume, mentioned above) in the paper, or in an appendix to it.*

A technical appendix has been added as supplementary material. This appendix includes specifics on the filters used and gives an idea over which frequency bands the filters were applied. Due to the nature of the project different cut-off frequencies in the band pass filtering were used dependent on source so the values quoted here are for a typical configuration.

*I would like to see more space devoted to the issues around accessible design overall, as this aspect of the project sets it apart from many art-science collaborations, and raises very important considerations in the transmission of both scientific and artistic/aesthetic information. Considerations around who our audiences are, and what are appropriate means of conveying ideas and information to them should be a first priority in this type of work, if we are to make inroads in communicating the relevance of both science and art to a wide audience. I commend the authors for raising this issue – but feel they could have addressed it more thoroughly, especially in relation to the user experience in the installation.*

We have made substantial revisions to the manuscript. Section 4 highlights accessibility throughout the project. Section 5 includes information on the curation and set up of hertz at each venue location

*It would also be of benefit to contextualize the project further; framing hertz in relation to both research in infrasound and in the context of contemporary sound art would allow readers to better situate the project's relevance to developments in both disciplines, and highlight the benefits of such collaborations. Examples of this work may be found at: Gupfinger, Ogawa, Sommerer, and Mignonneau (2009), Esquerro and Simon (2019),Sussman (2012), Hope (2009), Cranshaw (2014). In addition, there are other artists working with chladni plates; referencing their work would also assist in contextualizing the is aspect of the project, and strengthen the case for the relevance of this portion of the project here, and in future articles.*

In the revised version of this paper we have modified the introduction section to cite more references to show how the hertz artwork will sit within the broader context with other artworks that aim to make the intangible tangible. We have also included examples of other artworks including infrasound.

*I commend the authors for including commentary on their own experiences of working collaboratively, across disciplines. This is challenging work, and can only be truly successful if everyone involved approaches the work with openness, and a desire to learn and work in new ways. There is tremendous value in this approach to the explication of both complex scientific concepts and artistic creation alike, and much to be learned on both 'sides.'*

*Specific comments related to the above are listed by line number here:*

*Line 93 The relationship (if any) between the data collected by the CTBTO and the data collected through the INFRA 20 is not clearly stated here.*

The mention of the CTBTO was to give a broader context to why infrasound data was recorded originally and data from the CTBTO was not explicitly used in the artwork. Mention of the CTBTO has been removed.

*Line 65 Although you are not detailing this part of the project here, it is part of your documentation of installations, and the statistics on visitor interaction with the works(Page 8, line 250). As such, would be useful to readers to cite and/or refer to work in cymatics, perhaps in particular reference to contemporary art, to further explicate the notion of making the invisible visible. The work of both Nigel Stanford and Gary James Joynes come to mind.*

Although the Chladni plates are not the main subject of the paper we have cited the work of Nigel Stanford to give the interested reader more context if required.

*Line 120 -121 Could the sound emitted form the subwoofer also be felt physically? Worth noting one way or the other, as the secondary physical impact of the sound would contribute to the immersive quality. This seems to be the case given what you say in the next sentence about playing Dark Side of the Moon through the subwoofer & transducer in the first trial, and with respect to the subwoofer in the public iterations of the project. More detailed specifications for the subwoofer and transducer, and dB for both initial test and subsequent installations would be extremely helpful.*

Yes, the sound from the subwoofer could be felt physically. The dB was never explicitly measured nor calculated. The information about the power of the subwoofer and transducer will give some indicator to the loudness. Figure 7 has been added which shows a deaf participant feeling the motion of the air generated from the sub-woofer.

*Line 160 You could make more of the immediacy of the experience - it is an important factor in work of this nature that strives to connect people both emotionally and intellectually to natural phenomena such as this. This experience cannot be duplicated on the web, and cannot be simply listened to or watched: it needed people to be physically present. This becomes even more relevant in later iterations of the work, in which you draw on infrasound from the locations of presentation, where place and the experience of the work are inextricably linked.*

We have added in sections 2 and 5 descriptions of the project which highlight it was an experience that was specific to location and feel of the space

*Line 170 Was there feedback from the participants that was negative? Given the range of abilities in the audience for this, some may have had a negative experience; it would be useful to know this, what those less-positive responses were, and how they were factored in to further development of the project. For example, for some individuals with chronic pain and/or migraine and/or disabilities that affect balance, this installation may have been difficult to engage with, depending upon the volume or level of vibration physically experienced.*

Section 2.4 has been modified to ascertain if participants found the prototype installation unnerving or uncomfortable. Section 4 discusses how the hertz project focused on accessibility from initial discussions, venue choice, installation, right through to the design of interpretation materials. The feed back section has also been revised to provide a more thematic analysis.

*References Cited above:*

*Gupfinger, Reinhard & Ogawa, Hideaki & Sommerer, Christa & Mignonneau, Laurent.(2009). INTERACTIVE INFRASONIC ENVIRONMENT: A New Type of Sound Installation for Controlling Infrasound.*

*Ezquerro, L., and J. L. Simón. "Geomusic as a New Pedagogical and Outreach Re-source: Interpreting Geoheritage with All the Senses." Geoheritage 11, no. 3 (Septem-ber 1, 2019): 1187–98. https://doi.org/10.1007/s12371-019-00364-3.Sussman,M.*

*"HearingwithyourBody:Infrasound https://www.artpractical.com/feature/hearing_with_your_body_infrasound/# Accessed8 April 2020.Hope, Cat. "Infrasonic Music." Leonardo Music Journal 2009 Vol. 19, 51-56.*

*Hope, Cat. "Earth pulse: Vibrational data as artistic inspiration." Re:Live Media Arthistories 2009 Refereed Conference Proceedings (pp. 73-77), The University of Mel-bourne, 2009.Crawshaw, Alexis Story. "Towards Defining the Potential of Electroacoustic InfrasonicMusic." ICMC (2014).*

*Nigel Stanford. https://nigelstanford.com/Cymatics/Behind_the_Scenes.aspxGary James Jones. http://www.clinkersound.com/frequency-painting/?page_id=347Please also note the supplement to this comment:https://www.geosci-commun-discuss.net/gc-2020-9/gc-2020-9-RC3-supplement.pdfInteractive comment on Geosci. Commun. Discuss., https://doi.org/10.5194/gc-2020-9, 2020.C5*

**Referee 4:**

We thank the reviewer for their comments and respond to their comments below:

*Anonymous Referee #4*

*General comments:*

*This paper highlights a fascinating project that brings together science and art in a strong collaboration. It should prove interesting to scientists with an interest in public engagement with research, as well as artists looking to draw on science. However, I think there are some weaknesses in the paper as it is presented, which make it difficult to follow and detract from what is otherwise interesting work. It is lacking in clarity at times and I believe it would benefit from more detail at certain points.*

*Specific comments: There are clear connections made to interesting and relevant science content through the early stages of the project, during design and prototyping. The connection between research and the final installation is less clear, aside from the use of infrasound. High-lighting the scientific research content and how it was expressed in the main installation would substantially strengthen the connection to scientific questions, I think. I would have appreciated an explication of the context of the project among related artworks. Have any other art installations used infrasound or is this the first? Are there other works that have used vibration in a similar or different way? How does this installation relate to other experiential works that incorporate scientific data? Similarly, explaining what other public engagement or artistic projects exist around infrasound or ARISE 2 would have helped site this work in the relevant landscape.*

The introduction section has been amended to include references to similar artworks and their artists, with aims to contextualise the artwork in a broader context.

*Together, these would make much clearer the extent to which this work is novel. Occasionally, the paper refers to 'playing the infrasound' (e.g. line 186, 187). I think this is a little disingenuous. The processing is quite carefully described, but my understanding of it from this is that what is actually played is synthetically generated pinknoise, which is then processed according to the infrasound data. If my understanding is not quite right, then perhaps the section on the signal processing needs to be revisited. I wonder to what extent artistic licence was employed when creating the infrasound-scapes. Phrases such as 'This produced an effect that we felt was relatable to infrasound if we could hear it' suggest quite a lot, which in turn suggests a move away from the science. Perhaps the phrase 'keeping translatable authenticity' needs unpacking to clarify to what extent the experienced signals relate to the original infrasound signals. Relatedly, a flow diagram of sorts might help here (e.g. aroundline 152), to make clear exactly what the inputs, processes and outputs were for the prototypes and also for the final installation. Even after multiple readings of the paper, I'm still not sure I understand the relationship between the signals fed to the subwoofer and transducer – are they just the same?*

The author is correct. The artificially generated pink noise amplitude is modulated by the amplitude of the band pass filtered infrasound. Scientific license is added by setting the band pass filter bands based on the frequency domain of the detected infrasound which is shown in figures 3 & 4. This ensures that only the band pass filtered signal is used to modulate the pink noise. The modulated pink noise was then low passed filtered to only enable the low frequency parts of pink noise to be played via the PCs sound card to the subwoofer and transducer. To clarify this a technical diagram and flowchart are included in the technical appendix as part of the supplementary material

*The information on CTBTO stations and sensors is interesting, but I don't understand the connection between this and the project. Was data from these sensors used? Are the microbarometers used by CTBTO the same as the microbarometer used on this project? The connection needs to be made clear; or if there is not one, this (lines87-91) is probably extraneous and distracting information.*

The inclusion of the CTBTO was to include some context as to why and how infrasound is monitored across the globe to give the reader some additional background. The data detected by our sensor is effectively the same kind of data collected by the CTBTO sensors but isn't included in the project itself. Mention of the CTBTO has been removed from the manuscript

*Likewise, the reference to playing Pink Floyd through the system is confusing – did Pink Floyd use infrasound? Or was Pink Floyd's music used somewhere in the project? If there isn't any further connection, then I would suggest it is a distracting detail.*

It was a sound file used as a to initially test the system this has been stated in the text. It has been left in as it was a milestone for the authors in the initial construction of the prototype.

*A point that is made in passing, but that I think deserves much more attention, is that 'you had to be physically present in order to sense the frequencies, making it an immersive and experiential artwork'. The fact that the artwork could not be reproduced through audio or video recording marks it out as something special in a world that seems increasingly focused on engaging publics with research digitally. I think perhaps more could be made of this in the wider context of public engagement with research. On the other hand, Section 4.4 on Web and online presence comes across as rather weak. Simply stating the numbers of impressions gives no context and no conclusion. Can any analysis be done of who the Twitter followers were or who visited the website? Were they scientists? Artists? Funders? What were the most popular posts and why? How does this performance compare to similar websites or accounts? The weakness of simply stating figures is noted in the text, but if nothing further can be added to this section by way of analysis, I would consider removing it. As it stands, I think it detracts from the flow of the paper.*

We have removed the web presence section this and section 6 (the feedback section) has been refocused to look at all available feedback from across all venues to assess whether hertz met is aims to reconnect people with the Earth. A thematic analysis is also undertaken to attempt to understand this reconnection

*The assessment of feedback from the tour was also somewhat underwhelming. It seems largely to consist of sharing positive comments. This section would be much stronger if this was better contextualised. How many comments were received? How many of those were positive / negative? Can the feedback be analysed in more detail? The word cloud seems like a good start, but are there themes to be drawn out? A clearer explanation of how this feedback impacted the project would also be beneficial. On a different note, I don't think the description of the installation as "scary" needs to be considered negative, especially if part of the goal was to "re-establish links with the natural environment" including events that are "both majestic and alarming".*

Due to the nature of the tour the co-authors were only present at the first event and were not able to oversee the data collection at the other venues in person. Thus the feedback received was dependent on the venue in question, for example: At "We the Curious" the quality of the feedback received was quite good. However, Tramway's feedback did not capture the public's feedback and only that of the organisers.

Further to this feedback from the participants was entirely optional and the feedback cards left had little in the way of prompters. In hindsight it may have been better to devise 2 or 3 well defined questions to be asked on the exit of the exhibits.

Given the above highlighted issues with data quality we have rewritten the feedback section and perform a different analysis which would seek to answer, using the data available, Did participants feel more connected with the earth after interacting with the exhibits. This has been undertaken using a thematic approach as suggested by the reviewer

*Much is made of the artist-scientist relationship in this work, and to my mind this (Section 5) is the weakest section of the paper; I would consider substantially reducing or rewriting it with a much tighter focus. A substantial portion of the text is devoted to expounding stereotypes about the differences between how scientists work and how artists work. This struck me as rather lazy writing. There are no citations of studies or research that look at this question, and I wonder what the basis is for these wide-ranging assertions about what scientists "will" do. Furthermore, as this paragraph progresses, it seems to lose its line of argument, and it is not clear what point is intended. Moreover, I would be wary of suggesting that the different ways two particular people react to a particular event (see lines 305-309) is as a result of one being an artist and the other a scientist – this is not a strong conclusion.*

*Finally, the overall conclusion suggests that this collaboration "is a good model for future art science collaborations". To be more useful, I think the "model" in question needs further explanation. What was it they did that meant it worked especially well? What do other people need to know to be able to use the same model?*

This section has been re written, we have given the section more structure and have related our process through the project with current generalizations or other collaborations and have highlighted where these differ or are similar. We have also included how the project has improved or changed our working practices.

*There are a number of grammatical and punctuation errors throughout the text that need fixing. I think it could also do with the attention of a copy editor to re-phrase a few passages as some of the writing is a little stilted. Amending these would substantially improve the readability.*

A thorough proofread has been undertaken

*Technical corrections: There were a couple of names and phrases that I think need explaining in the text. A few words of context would save me looking it up and give me a better frame of reference.- Line 99, 'LT' is not explained – is this 'local time'?-*

This has been defined

*Line 165, what is the Attenborough Centre – is it an art space, a science space, a community space or something else?*

A line detailing the function of the Attenborough Centre has been added

*Line 211: What is 'We the Curious'?*

It is an educational science gallery We have added this in the revised manuscript

*There are some straightforward grammatical errors and misuses of punctuation-*

*Line 32: 'science technology engineering and maths' needs some commas-*

Commas have been added

*Line 36: 'one of those was, co-author' – unnecessary comma-*

This has been removed

*Line 41: '(Wilson 1969) see figure 1' – needs some punctuation-*

This has been changed

*Line 58: 'the star in turn fluctuates in brightness, satellites like Kepler' – probably fullstop, not comma-*

This sentence has been rephrased

*Line 59: 'transiting exoplanet survey satellite' – this is the name of a particular satellite,treat it as such-*

Lines 58 and 59 have been reworded

*Line 98-99: inconsistency with spaces before 'Hz'-*

This has been amended

*Line 150: 'synthetic generated pink noise. to ensure' – capital T on 'to'*

This has been amended

*Some sentences need re-phrasing, including-*

*Line 25: 'Technology further isolates the modern human from the natural environment in which we evolved increasingly being used as a filter through which we view the natural world.'-*

This sentence has been rephrased

*Line 57: 'Sound waves move through sun stars gaseous interior because of temperature changes'-*

This has been rephrased

*Line 106: 'The infrasonic signals produced by the Reading thunderstorms and the infrasonic signal from the aurora is 4 times smaller.' Four times smaller than what?-*

This has been rephrased to improve clarity between the infrasonic signatures recorded in figure 3 and 4

*Line 107: 'This shows that different phenomena produce have different infrasound signatures'*

This has been rephrased

[revised manuscript text omitted]